# Specific Neuroligin3–αNeurexin1 signaling regulates GABAergic synaptic function in mouse hippocampus

Motokazu Uchigashima[1,2], Kohtarou Konno[3], Emily Demchak[4], Amy Cheung[1], Takuya Watanabe[1], David G Keener[1], Manabu Abe[5], Timmy Le[1], Kenji Sakimura[5], Toshikuni Sasaoka[6], Takeshi Uemura[7,8], Yuka Imamura Kawasawa[4,9], Masahiko Watanabe[3], Kensuke Futai[1]*

[1]Brudnick Neuropsychiatric Research Institute, Department of Neurobiology, University of Massachusetts Medical School, Worcester, United States; [2]Department of Cellular Neuropathology, Brain Research Institute, Niigata University, Niigata, Japan; [3]Department of Anatomy, Faculty of Medicine, Hokkaido University, Sapporo, Japan; [4]Department of Biochemistry and Molecular Biology and Institute for Personalized Medicine, Pennsylvania State University College of Medicine, Hershey, United States; [5]Department of Animal Model Development, Brain Research Institute, Niigata University, Niigata, Japan; [6]Department of Comparative and Experimental Medicine, Brain Research Institute, Niigata University, Niigata, Japan; [7]Division of Gene Research, Research Center for Supports to Advanced Science, Shinshu University, Nagano, Japan; [8]Institute for Biomedical Sciences, Interdisciplinary Cluster for Cutting Edge Research, Shinshu University, Nagano, Japan; [9]Department of Pharmacology Pennsylvania State University College of Medicine, Hershey, United States

*For correspondence:
kensuke.futai@umassmed.edu

Competing interests: The authors declare that no competing interests exist.

**Abstract** Synapse formation and regulation require signaling interactions between pre- and postsynaptic proteins, notably cell adhesion molecules (CAMs). It has been proposed that the functions of neuroligins (Nlgns), postsynaptic CAMs, rely on the formation of trans-synaptic complexes with neurexins (Nrxns), presynaptic CAMs. Nlgn3 is a unique Nlgn isoform that localizes at both excitatory and inhibitory synapses. However, Nlgn3 function mediated via Nrxn interactions is unknown. Here we demonstrate that Nlgn3 localizes at postsynaptic sites apposing vesicular glutamate transporter 3-expressing (VGT3+) inhibitory terminals and regulates VGT3+ inhibitory interneuron-mediated synaptic transmission in mouse organotypic slice cultures. Gene expression analysis of interneurons revealed that the αNrxn1+AS4 splice isoform is highly expressed in VGT3+ interneurons as compared with other interneurons. Most importantly, postsynaptic Nlgn3 requires presynaptic αNrxn1+AS4 expressed in VGT3+ interneurons to regulate inhibitory synaptic transmission. Our results indicate that specific Nlgn–Nrxn signaling generates distinct functional properties at synapses.

## Introduction

In central synapses, cell adhesion molecules (CAMs) are major players in trans-synaptic interactions (*de Wit and Ghosh, 2016*) that serve a primary role in initiating synapse formation by directing contact between axonal and dendritic membranes. Emerging evidence suggests that trans-synaptic interactions are also important for synapse identity, function, plasticity, and maintenance (*Biederer et al., 2017*; *Campbell and Tyagarajan, 2019*; *Südhof, 2017*). Numerous CAM variants

exist due to large gene families and alternative splicing, generating a vast array of possible combinations of pre- and postsynaptic CAMs. Although some specific trans-synaptic interactions of CAMs have been reported to underlie distinct synaptic properties (*Chih et al., 2006*; *Fossati et al., 2019*; *Futai et al., 2013*), elucidating synaptic CAM complexes that dictate synapse identity and function remains a major challenge.

Four neuroligin (Nlgn) genes (*Nlgn1, Nlgn2, Nlgn3,* and *Nlgn4*) encode postsynaptic CAMs (Nlgn1, Nlgn2, Nlgn3, and Nlgn4) that contain extracellular cholinesterase-like domains and transmembrane and PDZ-binding motif-containing intracellular domains. Each Nlgn protein has a distinct pattern of subcellular localization at excitatory, inhibitory, dopaminergic, and cholinergic synapses (*Song et al., 1999*; *Takács et al., 2013*; *Uchigashima et al., 2016*; *Varoqueaux et al., 2004*). Interestingly, Nlgn3 is the only Nlgn isoform localized at both excitatory and inhibitory synapses (*Baudouin et al., 2012*; *Budreck and Scheiffele, 2007*; *Uchigashima et al., 2020*), regulating their synaptic functions (*Etherton et al., 2011*; *Földy et al., 2013*; *Horn and Nicoll, 2018*; *Shipman et al., 2011*; *Tabuchi et al., 2007*). However, the trans-synaptic framework that dictates Nlgn3 function is poorly understood.

Neurexins (Nrxns) are presynaptic CAMs produced from three genes (*Nrxn1*, *Nrxn2*, and *Nrxn3*) that are transcribed from different promoters as longer alpha (*αNrxn1–3*), shorter beta (*βNrxn1–3*), and *Nrxn1*-specific gamma (*γNrxn1*) isoforms (*Sterky et al., 2017*; *Tabuchi and Südhof, 2002*), and serve as the sole presynaptic binding partners for Nlgns. Each *Nrxn* gene has six alternative splicing sites, named AS1–AS6, resulting in thousands of potential *Nrxn* splice isoforms (*Górecki et al., 1999*; *Missler et al., 1998*; *Püschel and Betz, 1995*; *Schreiner et al., 2014*; *Treutlein et al., 2014*; *Ullrich et al., 1995*). Unique transcription patterns of *Nrxns* have been observed in hippocampal interneurons, suggesting that Nrxn proteins may determine the properties of GABAergic synapses in an input cell-dependent manner (*Fuccillo et al., 2015*).

Nrxn–Nlgn interactions depend on Nrxn protein length (long form [α] vs short form [β]), splice insertions at AS4 of Nrxns, and splice insertions of Nlgns. For example, Nlgn1 splice variants that have splice insertions at site B have higher binding affinities for βNrxn1-AS4 (βNrxn1 lacking alternative splice insertion at AS4) than for βNrxn1+AS4 (containing an alternative splice insertion at AS4) (*Boucard et al., 2005*; *Koehnke et al., 2010*; *Reissner et al., 2008*). However, it is largely unknown which Nrxn–Nlgn combination defines specific synapse functionality. We recently found that Nlgn3Δ, which lacks both of the A1 and A2 alternative splice insertions, is the major Nlgn3 splice isoform expressed in hippocampal CA1 pyramidal neurons and regulates both excitatory and inhibitory synaptic transmission (*Uchigashima et al., 2020*). However, to the best of our knowledge, the synapses at which Nlgn3Δ interacts with presynaptic Nrxn isoform(s) have not been identified.

Interneurons exhibit extraordinary morphological, physiological, and molecular diversity in the cortex and hippocampus (*Klausberger and Somogyi, 2008*; *Markram et al., 2004*; *Pelkey et al., 2017*; *Somogyi and Klausberger, 2005*). Indeed, there are over 20 classes of inhibitory interneurons in the CA1 area based on molecular markers, action potential (AP) firing patterns, and morphology (*Klausberger and Somogyi, 2008*; *Pelkey et al., 2017*). Among them, interneurons expressing parvalbumin (Pv+), somatostatin (Sst+), and cholecystokinin (Cck+) display different morphologies, excitability, and synaptic functions. However, the molecular mechanisms underlying their diversity are unknown.

In the present study, we show that Nlgn3 is selectively enriched at vesicular glutamate transporter 3-expressing (VGT3+) Cck+ inhibitory terminals in the hippocampal CA1 area. Gain-of-function and loss-of-function studies revealed that Nlgn3 regulates VGT3+ interneuron-mediated inhibitory synaptic transmission. Importantly, the effect of Nlgn3 on VGT3+ synapses was hampered by the deletion of all *Nrxn* genes in VGT3+ interneurons and rescued by the selective expression of αNrxn1+AS4 in VGT3+ interneurons. These results suggest that the trans-synaptic interaction between αNrxn1+AS4 and Nlgn3 underlies the input cell-dependent control of VGT3+ GABAergic synapses in the hippocampus.

## Results

### Nlgn3 is enriched at VGT3+ GABAergic synapses in the hippocampal CA1 region

In a recent study, we demonstrated that Nlgn3 localizes at and regulates both inhibitory and excitatory synapses in the hippocampal CA1 area (*Uchigashima et al., 2020*). However, the distribution of Nlgn3 at different types of inhibitory synapses has not yet been addressed. Therefore, we first examined which GABAergic inhibitory synapses express Nlgn3 in the CA1 area by immunohistochemistry. Cck+, Pv+, and Sst+ interneurons are the primary inhibitory neurons in the hippocampus. Moreover, the cell bodies and dendritic shafts of CA1 pyramidal cells are targeted by Cck+ and Pv+, and Sst+ interneurons, respectively (*Pelkey et al., 2017*). Our Nlgn3 antibody with specific immunoreactivity was validated in Nlgn3 KO brain (*Figure 1—figure supplement 1A and B*) and displayed a typical membrane protein distribution pattern in the hippocampus, as we recently reported (*Figure 1A*; *Uchigashima et al., 2020*). Inhibitory synapses expressing Nlgn3 were identified by co-localization of Nlgn3 signals with vesicular inhibitory amino acid transporter (VIAAT) signals. Four different inhibitory axons/terminals were visualized by anti-VGT3 and -CB1 (markers for Cck+ interneurons) (*Früh et al., 2016*), -Pv, and -Sst antibodies. We found that signal intensities for Nlgn3 were considerably high at GABAergic synapses co-labeled with VGT3 or CB1 (*Figure 1B,C, and F*) and low or at approximately noise levels at those co-labeled with Pv (*Figure 1D and F*) or Sst (*Figure 1E and F*) in the CA1 region. Noise levels were obtained from images that lacked true close apposition of signals for Nlgn3 and synaptic markers observed by rotating the Nlgn3 channel 90° (*Figure 1F* and *Figure 1—figure supplement 1G*). Moreover, co-localization of Nlgn3 signals with these markers demonstrated similar findings (*Figure 1G*). Therefore, these data strongly suggest that Nlgn3 is preferentially recruited to Cck+ GABAergic synapses, but not to Pv+ or Sst+ inhibitory synapses.

### Nlgn3 regulates inhibitory synaptic transmission at VGT3+ GABAergic synapses

To determine whether Nlgn3 has specific roles at VGT3+ inhibitory synapses, we assessed the effect of overexpressing Nlgn3Δ, the major Nlgn3 splice isoform expressed in CA1 pyramidal neurons (*Uchigashima et al., 2020*), on input-specific inhibitory transmission. To distinguish a subset of GABAergic synapses and evoke cell-specific synaptic transmission, we generated three cell type-specific fluorescent lines by crossing VGT3-Cre, Sst-Cre, or Pv-Cre with a TdTomato (RFP) reporter line, producing respectively, VGT3/RFP, Sst/RFP, and Pv/RFP mouse lines. TdTomato-expressing cells in each of the three fluorescent mouse lines were distributed in the CA1 in a layer-dependent manner (*Figure 2—figure supplement 1*). We evaluated the effect of Nlgn3Δ overexpression (OE) on unitary inhibitory postsynaptic currents (uIPSCs) by triple whole-cell recordings using organotypic slice cultures from each mouse line. Two to three days after transfection of Nlgn3Δ or enhanced green fluorescent protein (EGFP) control by biolistic gene gun, current- and voltage-clamp recordings were conducted from a presynaptic RFP+ interneuron and postsynaptic EGFP or EGFP/Nlgn3Δ-positive and -negative postsynaptic pyramidal neurons, respectively (*Figure 2A*). RFP interneurons expressing VGT3 in the pyramidal cell layer and stratum (st.) oriens and radiatum, Pv in the st. pyramidale and Sst in the st. oriens, were chosen (*Figure 2—figure supplement 1*). uIPSCs were evoked by inducing APs in RFP+ neurons. The amplitude and paired-pulse ratio (PPR), monitoring release probability, of uIPSCs and synaptic connectivity were compared between Nlgn3Δ-transfected and -untransfected neurons (*Figure 2B–E*). Importantly, VGT3+ inhibitory interneurons displayed clear potentiation of uIPSCs onto CA1 pyramidal neurons overexpressing Nlgn3Δ (*Figure 2C*). Paired APs of VGT3+ neurons with short intervals (50 ms) induced paired-pulse depression (PPD) of uIPSCs. Nlgn3Δ displayed reduced PPD compared with untransfected neurons, consistent with previous work (*Futai et al., 2007*; *Shipman et al., 2011*; *Uchigashima et al., 2020*; *Figure 2D*). As PPR inversely correlates with presynaptic release probability, these results suggest that Nlgn3Δ OE can facilitate presynaptic GABA release. In contrast, Nlgn3Δ OE reduced uIPSCs in Pv+ inhibitory synaptic transmission, but had no effect on PPR as reported previously (*Figure 3A–D*; *Horn and Nicoll, 2018*). Lastly, Nlgn3Δ OE did not alter uIPSCs or PPR mediated by Sst+ interneurons (*Figure 3E–H*). No effect of biolistic transfection with EGFP alone was found on uIPSC amplitude, PPR, or

**Figure 1.** Nlgn3 is predominantly expressed in VGT3+ and CB1+ inhibitory synapses. (A) Single immunofluorescence of Nlgn3 in the hippocampal CA1 region shows a basket-like clustering of Nlgn3 immunofluorescent signals around the somata of pyramidal cells. Py, st. pyramidale; Ra, st. radiatum. (B–E) Triple immunofluorescence for Nlgn3, VIAAT, and interneuron markers: VGT3 (B), CB1 (C), Pv (D), and Sst (E). The boxed area in low magnification images is enlarged in lower panels. Arrowheads indicate Nlgn3 immunofluorescent puncta associated with (yellow) or distant from (white) interneuron markers. (F and G) Summary of the relative intensi[■■■■■■■■]frequency (G) of Nlgn3 immunofluorescent signals at different inhibitory synapses. Plots are obtained from each synapse for i[■■■■■■■■]r co-localizations (G). Noise levels for the intensity and co-localization were obtained from images with the Nlgn3 chann[■■■■■■■] rotation). ***p<0.001; n.s. not significant; One-way ANOVA with Sidak's post hoc test. Bars on each column represent mean ± SEM. Scale bars, 100 μm (A) and 2 μm (B–E).

The online version of this article includes the following figure supplement(s) for figure 1:

**Figure supplement 1.** Validation of anti-Nlgn3 antibody.

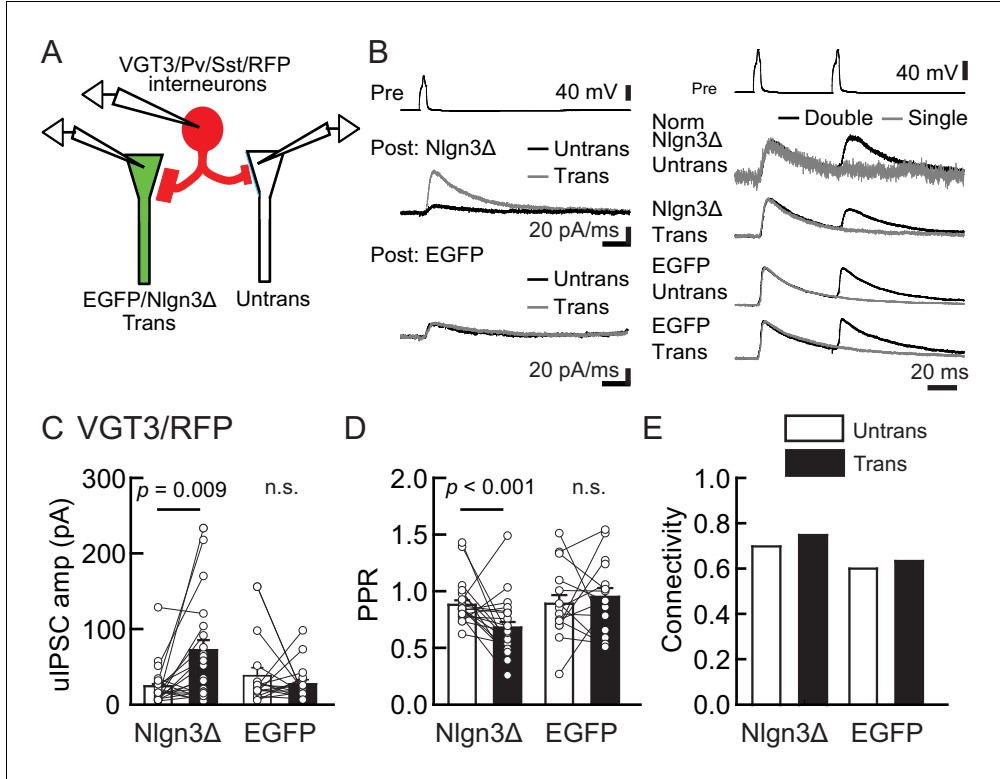

**Figure 2.** Nlgn3Δ overexpression (OE) specifically potentiates VGT3+ interneuron-mediated unitary synaptic transmission. The effects of OE of Nlgn3Δ isoform or enhanced green fluorescent protein (EGFP, control) in hippocampal CA1 pyramidal neurons on inhibitory inputs mediated by VGT3+ inhibitory interneurons. (**A**) Configuration of the triple whole-cell recording for VGT3+, Pv+, and Sst+ interneurons forming synapses with pyramidal neurons. (**B**) Sample traces of unitary inhibitory postsynaptic currents (uIPSCs). Left top, averaged sample traces of a single presynaptic action potential (AP) evoked in a VGT3+ interneuron. Left middle and bottom, superimposed averaged sample uIPSC traces (Untrans: black; trans: dark gray) induced by an AP. Right, superimposed averaged sample traces of uIPSCs evoked by single (dark gray) and double (black) APs in VGT3+ interneurons. uIPSCs are normalized to the first amplitude. Because the first uIPSC overlaps with the second uIPSC, to accurately measure the amplitude of the second IPSC, we 'cancelled' the first uIPSC by subtracting the traces receiving a single pulse (gray) from those receiving a paired pulse (black), both normalized to the first response. The amplitude (**C**) and paired-pulse ratio (PPR) (**D**) of uIPSCs were plotted for each pair of transfected (Trans) and neighboring untransfected (Untrans) cells (open symbols). Bar graphs indicate mean ± SEM. (**E**) Synaptic connectivity between presynaptic inhibitory interneuron and postsynaptic untransfected (open bars) or transfected (black) pyramidal neurons. Numbers of cell pairs: Nlgn3Δ or EGFP at VGT3+ synapses (42 pairs/23 mice and 30/21). The number of tested slice cultures is the same as that of cell pairs. n.s., not significant. Mann–Whitney U-test.

The online version of this article includes the following figure supplement(s) for figure 2:

**Figure supplement 1.** Validation of TdTomato expression in three different cell type-specific fluorescent lines.

---

connection probability at Pv+ and Sst+ inhibitory synapses. The above results strongly suggest that Nlgn3 modifies inhibitory synaptic function depending on the type of presynaptic interneuron with which it interacts. Interestingly, Nlgn3Δ OE did not increase synaptic connectivity (*Figure 2E*), suggesting that (i) Nlgn3 regulates pre-existing inhibitory inputs on postsynaptic neurons and/or (ii) postsynaptic Nlgn3Δ OE is not sufficient to induce new synapse formation.

Overexpression of postsynaptic Nlgn3A2, a Nlgn3 splice isoform including the A2 cassette, in CA1 pyramidal neurons has been reported to differentially regulate Pv+ and Sst+ inhibitory synapses (*Horn and Nicoll, 2018*). Horn and Nicoll reported that human Nlgn3A2 OE reduces Pv+ and increases Sst+ inhibitory synaptic transmission, which is inconsistent with our findings in Sst+ synapses (*Figure 3F*). This suggests that the signaling interaction between the input neuron and different Nlgn3 splice isoforms may generate distinct inhibitory regulatory mechanisms. In addition, we also

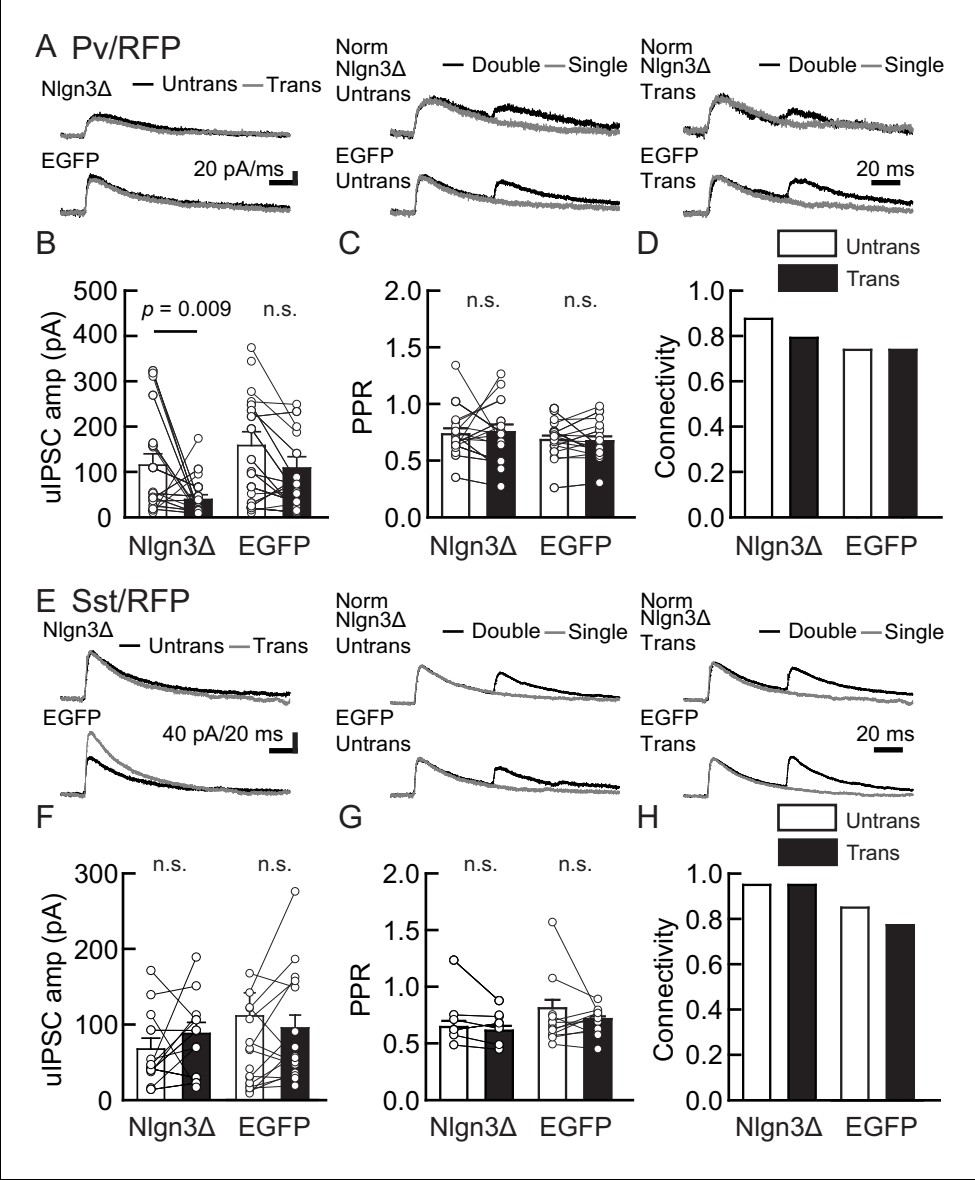

**Figure 3.** Nlgn3Δ overexpression (OE) does not increase PV+ and Sst+ interneuron-mediated unitary synaptic transmission. The effects of OE of Nlgn3Δ isoform or enhanced green fluorescent protein (EGFP; control) in hippocampal CA1 pyramidal neurons were compared between inhibitory inputs mediated by Pv+ (**A–D**) and Sst+ (**E–H**) inhibitory interneurons. (**A and E**) Sample traces of unitary inhibitory postsynaptic currents (uIPSCs). Left, superimposed averaged sample uIPSC traces (Untrans: black; trans: dark gray) induced by an action potential (AP). Middle and right, superimposed averaged sample traces of uIPSCs evoked by single (dark gray) and double (black) APs in Pv+ and Sst+ interneurons. uIPSCs are normalized to the first amplitude. The amplitude (**B and F**) and paired-pulse ratio (PPR) (**C and G**) of uIPSCs were plotted for each pair of transfected (Trans) and neighboring untransfected (Untrans) cells (open symbols). Bar graphs indicate mean ± SEM. (**D and H**) Synaptic connectivity between presynaptic inhibitory interneuron and postsynaptic untransfected (open bars) or transfected (black) pyramidal neurons. Numbers of cell pairs: Nlgn3Δ or EGFP at Pv+ synapse (33 pairs/19 mice and 27/20) and Sst+ synapses (20/11 and 26/15). The number of tested slice cultures is the same as that of cell pairs. n.s., not significant. Mann–Whitney U-test.

The online version of this article includes the following figure supplement(s) for figure 3:

**Figure supplement 1.** Nlgn3A2 overexpression (OE) does not increase VGT3+, Pv+, and Sst+ interneuron-mediated unitary synaptic transmission.

reported that Nlgn3A2 OE increases evoked inhibitory synaptic transmission in the CA1 region (*Uchigashima et al., 2020*). Therefore, we tested the effect of mouse Nlgn3A2 OE on inhibitory synaptic transmission at Pv+, Sst+, and VGT3+ synapses (*Figure 3—figure supplement 1*). To our surprise, mouse Nlgn3A2 did not potentiate VGT3+ and Sst+ inhibitory synapses (*Figure 3—figure supplement 1A–H*). Nlgn3A2 OE reduced uIPSCs at Pv+ synapses, like Nlgn3Δ (*Figure 3I–L*). These findings suggest that Nlgn3A2 regulates inhibitory synapses through different interneuron type(s).

## Nlgn3 knockdown reduces VGT3+ inhibitory synaptic transmission in the hippocampal CA1 region

We next tested the impact of acute Nlgn3 knockdown (KD) on inhibitory synaptic transmission. Organotypic slice cultures prepared from C57BL/6J mice were biolistically transfected with shRNA against Nlgn3 (shNlgn3#1), which exhibits over 90% KD efficiency specific to Nlgn3 isoforms (*Figure 4—figure supplement 1A*), or control shRNA (shCntl). Transfection was performed at days in vitro (DIV) 2 and recordings were performed 7–9 days later to measure inhibitory synaptic transmission mediated by three different synaptic inputs (*Figure 4*). Compared with untransfected neurons, shNlgn3#1-transfected neurons displayed reduced uIPSC amplitudes mediated by VGT3+ interneurons (*Figure 4A–D*). In contrast, uIPSC amplitudes were affected in neither Pv+ (*Figure 4E–H*) nor Sst + (*Figure 4I–L*) neurons. Another Nlgn3 KD shRNA, shNlgn3#2, also reduced uIPSC amplitudes at VGT3+ inhibitory synapses (*Figure 4—figure supplement 1B–F*), ruling out off-target effects of shRNAs. Neurons transfected with shCntl displayed uIPSC amplitudes comparable to untransfected neurons. No significant changes were detected in PPR (*Figure 4C,G, and K*) and connection probability (*Figure 4D,H, and L*) between transfected and untransfected neurons. Taken together, these data suggest that Nlgn3 is required for synaptic transmission specifically at VGT3+ GABAergic synapses in an input cell-dependent manner.

## Lack of *Nrxn* genes in presynaptic VGT3+ neurons abolishes the effect of postsynaptic Nlgn3Δ OE

Our results suggest that Nlgn3 is preferentially located at VGT3+ inhibitory synapses and regulates inhibitory synaptic transmission. Postsynaptic Nlgns couple with presynaptic Nrxns to form trans-synaptic protein complexes that regulate synapse formation and function (*Südhof, 2017*). Does input-specific Nlgn3Δ (*Figure 2C*) require presynaptic Nrxn proteins? To address this question, we generated VGT3 neuron-specific *Nrxn* triple knockout (TKO) with TdTomato reporter gene mouse line (*Nrxn1/2/3$^{f/f}$*/VGT3-Cre/TdTomato: NrxnTKO/VGT3/RFP). This mouse line is fertile with KO of *Nrxn1, 2,* and *3* specifically in TdTomato-positive VGT3+ neurons (*Figure 5A*). First, we assessed the impact of Nrxn TKO on VGT3+ synaptic transmission. Pre- and postsynaptic dual whole-cell recordings were performed between RFP-positive VGT3+ interneurons in CA1 st. pyramidale and nearby CA1 pyramidal neurons (*Figure 5B–E*). NrxnTKO/VGT3/RFP mice displayed reduced uIPSC, PPR at 25 ms inter-pulse interval, and connectivity compared with wild-type (WT) VGT3/RFP mice (*Figure 5C–E*). Intrinsic excitability was comparable between WT and NrxnTKO VGT3+ neurons (*Figure 5—figure supplement 1*). These results suggest that Nrxns in VGT3+ interneurons regulate synaptic transmission without changing intrinsic membrane properties. Next, we transfected Nlgn3Δ in NrxnTKO/VGT3/RFP slice cultures and performed triple whole-cell recordings as described above using VGT3/RFP slice cultures (*Figure 2B–E*). VGT3+ interneurons lacking Nrxns induced synaptic release regardless of Nlgn3Δ gene transfection, indicating that presynaptic Nrxns in VGT3+ interneurons are not essential for synaptogenesis. Importantly, we observed no enhancement of uIPSC amplitude in Nlgn3-overexpressed neurons compared with untransfected pyramidal neurons (*Figure 5F–I*). These results strongly suggest that presynaptic Nrxn proteins are necessary for regulating inhibitory synaptic transmission through postsynaptic Nlgn3.

## $\alpha$*Nrxn1* and $\beta$*Nrxn3* mRNAs are highly expressed in VGT3+ interneurons

Our results above clearly suggest that presynaptic Nrxn proteins are important for the function of postsynaptic Nlgn3Δ. We therefore hypothesized that *Nrxn* isoforms highly expressed in VGT3+ interneurons functionally couple with postsynaptic Nlgn3Δ. To address this hypothesis, we examined the mRNA expression patterns of α and β isoforms of *Nrxn1–3* in hippocampal CA1 interneurons by

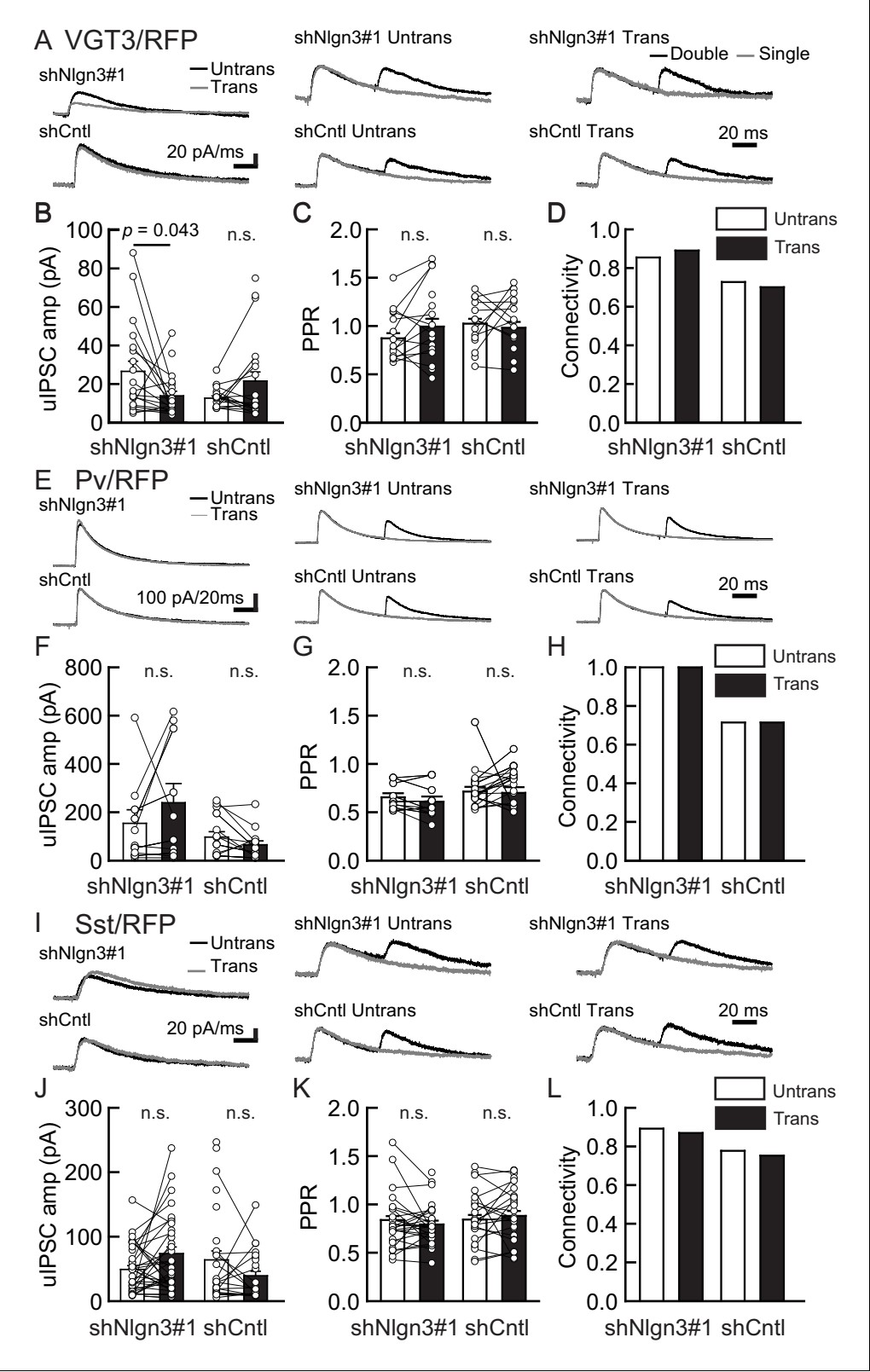

**Figure 4.** Endogenous Nlgn3 specifically regulates VGT3+, but not Pv+ and Sst+, interneuron-mediated unitary synaptic transmission. The effects of shNlgn3 or shCntl expression in hippocampal CA1 pyramidal neurons were compared between three different inhibitory inputs mediated by VGT3+ (**A–D**), Pv+ (**E–H**), and Sst+ (**I–L**) inhibitory interneurons. (**A, E, and I**) Sample traces of unitary inhibitory postsynaptic currents (uIPSCs). Left, superimposed

*Figure 4 continued*

averaged sample traces of uIPSC (Untrans: black; trans: dark gray) induced by an action potential (AP). Middle and right, superimposed averaged sample traces of uIPSCs evoked by single (dark gray) and double (black) APs in VGT3+ (A), Pv+ (E), and Sst+ (I) interneurons. uIPSCs are normalized to the first amplitude. The amplitude (B, F, and J) and paired-pulse ratio (PPR) (C, G, and K) of uIPSCs were plotted for each pair of transfected (Trans) and neighboring untransfected (Untrans) cells (open symbols). Bar graphs indicate mean ± SEM. (D, H, and L) Synaptic connectivity between presynaptic inhibitory interneuron and postsynaptic untransfected (open bars) or transfected (black) pyramidal neurons. Numbers of cell pairs: shNlgn3 or shCntl at VGT3+ synapses (28 pairs/15 mice and 37/17), at Pv+ synapse (12/4 and 16/5), and Sst+ synapses (26/11 and 21/10). The number of tested slice cultures is the same as that of cell pairs. n.s., not significant. Mann–Whitney U-test.

The online version of this article includes the following figure supplement(s) for figure 4:

**Figure supplement 1.** Nlgn3 shRNAs specifically knockdown Nlgn3 protein and regulate VGT3+ inhibitory synaptic transmission.

fluorescent in situ hybridization (FISH). The specificities of cRNA probes for 6 *Nrxn* isoforms were validated recently (*Uchigashima et al., 2019*). mRNAs encoding all *Nrxn* isoforms, except *βNrxn1*, were detected not only in the st. pyramidale but also in scattered cells across all layers (*Figure 6— figure supplement 1A–F*). Importantly, *αNrxn1* and *βNrxn3* mRNAs appeared to be enriched in scattered cells within the st. radiatum or pyramidale (arrows in *Figure 6A–C and G–I*), where VGT3+ interneurons are dominantly distributed compared with Pv+ and Sst+ interneurons (*Pelkey et al., 2017*). Double FISH signals were twice as strong for *αNrxn1* and *βNrxn3* mRNAs in VGT3+ (*Figure 6A,G, and J*) interneurons than in Pv+ (*Figure 6B,H, and J*) and Sst+ (*Figure 6C,I, and J*) interneurons. In contrast, there were no differences in the signal intensities for the remaining *Nrxn* isoforms between VGT3+ and other interneurons (*Figure 6D–F and J* and *Figure 6—figure supplement 1G–P*). These findings suggest that *αNrxn1* and *βNrxn3* mRNAs are highly expressed in VGT3 + interneurons compared with Pv+ or Sst+ interneurons.

## Expression profiles of *Nrxn* splice isoforms in VGT3+, Pv+, and Sst+ interneurons

Single-cell RNA sequencing was performed to elucidate splice variant expression of *Nrxn* isoforms in VGT3+, Pv+, and Sst+ interneurons. We harvested cytosol from five VGT3/, Pv/, and Sst/RFP neurons through whole-cell glass electrodes and performed single-cell deep RNA-seq. The t-SNE plot indicates that the genome-wide transcriptomes of the Pv+ and Sst+ interneurons (five cells each, denoted as MU_Pv.1–5 and MU_Sst.1–5) were clustered together and close to that of adult Pv+ and Sst+ neurons, respectively, derived from the hippocampal single-cell RNA-seq dataset in the Allen Brain Map Cell Types Database (Mouse – Hippocampus dataset; *Figure 7A* and *Figure 7—source data 1*). In contrast, the VGT3+ interneurons were more sparsely distributed near the clusters of Sncg+ and Sst+ interneurons. The expression of *Nrxn* genes in these cells was similar to that of GABAergic neurons (*Figure 7—figure supplement 1* and *Figure 7—source data 2*). The quantification of *Nrxn* genes indicates that the expression of Nrxn3 is dominant in these three types of interneurons (*Figure 7B* and *Figure 7—figure supplement 1*). We then compared the expression of *Nrxn* splice isoforms in each *Nrxn* gene. Given that the insertion of AS4 determines the binding of many Nrxn protein binding partners including Nlgns (*Südhof, 2008*; *Südhof, 2017*), we quantified the expression of *Nrxn* isoforms with or without AS4 insertion. Twelve *Nrxn* splice isoforms, *αNrxn1 +AS4*, *αNrxn1-AS4*, *αNrxn2+AS4*, *αNrxn2-AS4*, *αNrxn3+AS4*, *αNrxn3-AS4*, *βNrxn1+AS4*, *βNrxn1-AS4*, *βNrxn2+AS4*, *βNrxn2-AS4*, *βNrxn3+AS4*, and *βNrxn3-AS4* were manually modified (*Figure 7C–E* and *Table 1*), and their expression was compared. Among *Nrxn1* splice isoforms, *αNrxn1+AS4* was consistently detected as the sole *αNrxn1* gene expressed in the three interneurons and highly expressed in VGT3+ interneurons compared with Pv+ and Sst+ interneurons (*Figure 7C*). *βNrxn2+AS4* was the only confirmed *Nrxn2* gene expressed in the three interneurons but its expression was much lower than that of *Nrxn1* and *3* (*Figure 7D*). *αNrxn3+AS4* and *βNrxn3-AS4* were the two major *Nrxn3* genes expressed in the three interneurons, and *αNrxn3+AS4* expression was highest in VGT3+ interneurons compared with other interneurons (*Figure 7E*). Our two gene expression assays, double FISH and single-cell RNA-seq, suggest that *αNrxn1* and *βNrxn3* are the major *Nrxn* genes expressed in VGT3+ interneurons compared with Pv+ and Sst+ interneurons (*Figure 6J*), and

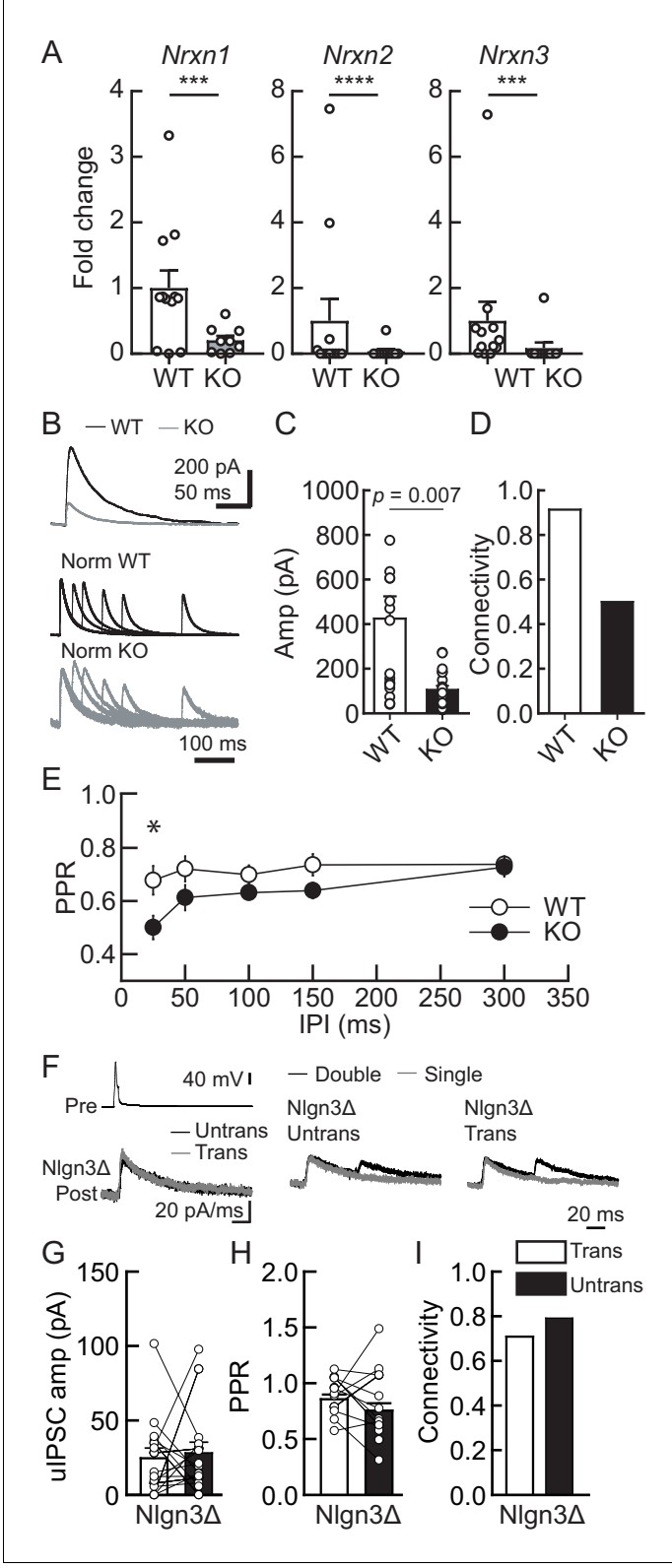

**Figure 5.** Lack of presynaptic Nrxns reduces inhibitory synaptic transmission and abolishes the potentiation effect of Nlgn3Δ in VGT3+ inhibitory synaptic transmission. Effects of Nlgn3Δ isoform overexpression (OE) in hippocampal CA1 pyramidal neurons in the absence of presynaptic Nrxn input in VGT3+ interneurons. (**A**) Validation of the NrxnTKO/VGT3/RFP mouse line. Expression of *Nrxn* genes were compared in TdTomato-positive VGT3+ neurons in organotypic hippocampal slice cultures prepared from wild-type (WT) (VGT3/RFP) and

*Figure 5 continued on next page*

*Figure 5 continued*

knockout (KO) (NrxnTKO/VGT3/RFP) mice. qPCRs against *Nrxn 1, 2, 3* and *Gapdh* (internal control) were performed for single-cell cDNA libraries prepared from TdTomato-positive neurons. Number of neurons: WT (N = 9, three mice) and KO (7, 2). ***p<0.001, ****p<0.0001 (Student's *t*-test). (**B–E**) Effect of Nrxn triple KO on VGT3+ interneuron-mediated inhibitory synaptic transmission. (**B**) Left top, averaged sample traces of a single presynaptic action potential (AP) evoked in a NrxnTKO VGT3+ interneuron. Left bottom, superimposed averaged sample traces of unitary inhibitory postsynaptic current (uIPSC) (Untrans: black; trans: dark gray) induced by an AP. Right, superimposed averaged sample traces of uIPSCs evoked by single (dark gray) and double (black) APs in NrxnTKO VGT3+ interneurons. Summary of uIPSC amplitude (**C**), paired-pulse ratio (PPR) (**D**), and connectivity (**E**). Number of cell pairs: WT (N = 23, six mice) and KO (35, 7). *p<0.05 (two-way ANOVA with Sidak's post hoc test). (**F–I**) Effect of Nlgn3Δ OE on NrxnTKO VGT3+ interneuron-mediated inhibitory synaptic transmission. (**F**) Left top, averaged sample traces of a single presynaptic AP evoked in a NrxnTKO VGT3+ interneuron. Left bottom, superimposed averaged sample traces of uIPSC (Untrans: black; trans: dark gray) induced by an AP. Right, superimposed averaged sample traces of uIPSCs evoked by single (dark gray) and double (black) APs in NrxnTKO VGT3+ interneurons. Summary of uIPSC amplitude (**G**), PPR (**H**), and connectivity (**I**). Open circles connected with bars represent individual pairs of cells (**C, D, G, and H**). Bar graphs indicate mean ± SEM. N = 18 cell pairs (three mice). The number of tested slice cultures is the same as that of cell pairs. Mann–Whitney U-test.

The online version of this article includes the following figure supplement(s) for figure 5:

**Figure supplement 1.** Membrane excitability is not altered in VGT3+ interneurons in NrxnTKO/VGT3/RFP mice.

---

αNrxn1+AS4, αNrxn3+AS4, and βNrxn3-AS4 are *Nrxn* splice isoforms highly expressed in VGT3+ interneurons (*Figure 7C and E*). Therefore, αNrxn1+AS4, αNrxn3+AS4, and βNrxn3-AS4 are the unique *Nrxn* genes expressed in VGT3+ interneurons compared with Pv+ and Sst+ interneurons.

## Presynaptic αNrxn1+AS4 couples with postsynaptic Nlgn3Δ to regulate inhibitory function

Our electrophysiology results using NrxnTKO/VGT3/RFP mice indicate that Nrxn proteins are important for postsynaptic Nlgn3Δ function. We sought to determine which Nrxn isoform(s) is the functional partner(s) of Nlgn3Δ at VGT3+ inhibitory synapses. Does VGT3+ interneuron-dominant *Nrxns*, αNrxn1, and βNrxn3 interact with postsynaptic Nlgn3Δ to modulate inhibitory synaptic function? To address this, we performed a rescue approach by expressing specific Nrxn isoforms in NrxnTKO/VGT3/RFP neurons. We expressed Nrxn with tag-blue fluorescent protein (tag-BFP) and Nlgn3Δ with EGFP in VGT3/RFP and pyramidal neurons, respectively, in NrxnTKO/VGT3/RFP slice cultures using our recently developed electroporation technique (*Keener et al., 2020a*; *Keener et al., 2020b*), and performed triple whole-cell recordings from untransfected and tag-BFP/Nrxn-transfected VGT3/RFP neurons, and GFP/Nlgn3Δ-transfected pyramidal neurons (*Figure 8*). We first identified two neighboring VGT3/RFP neurons in the hippocampal st. radiatum and oriens regions and electroporated tag-BFP/Nrxn plasmids into one of the cells, while GFP/Nlgn3Δ was electroporated into pyramidal neurons near the electroporated VGT3/RFP neuron. Given that splice insertion at site four in Nrxns regulates binding with postsynaptic Nlgns (*Südhof, 2008*; *Südhof, 2017*), we transfected four *Nrxn* splice isoforms, αNrxn1+AS4, αNrxn1-AS4, βNrxn3+AS4, or βNrxn3-AS4, together with tag-BFP into VGT3/RFP neurons in CA1 st. radiatum and pyramidale. Two to three days after electroporation, we performed triple whole-cell recording from *Nrxn*-transfected presynaptic interneurons (Tdtomato- and tag-BFP-positive) located in close proximity to untransfected VGT3/RFP (*Figure 8A*) and Nlgn3Δ-transfected (EGFP-positive) postsynaptic pyramidal neurons. Cell pairs overexpressing αNrxn1+AS4 and Nlgn3Δ in presynaptic VGT3/RFP and postsynaptic pyramidal neurons, respectively, displayed significant enhancement of uIPSC with 100% connectivity (*Figure 8B,C, and E*). In contrast, presynaptic αNrxn1-AS4/tag-BFP and postsynaptic GFP/Nlgn3Δ neuron pairs showed no enhancement of uIPSC compared with control cell pairs, suggesting that Nlgn3Δ specifically couples with αNrxn1+AS4 to regulate inhibitory synaptic function at VGT3+ synapses. Transfection of tag-BFP into VGT3/RFP neurons did not alter inhibitory synaptic transmission. Next, we tested βNrxn3-Nlgn3Δ synaptic *signals* on VGT3+ inhibitory synaptic transmission (*Figure 9A–D*). In contrast with αNrxn1–Nlgn3Δ signals, neither βNrxn3+AS4 nor βNrxn3-AS4 transfection showed detectable changes in uIPSCs compared with untransfected VGT3/RFP neurons, indicating that βNrxn3 is not important for Nlgn3-mediated inhibitory synaptic function. Interestingly, βNrxn3-AS4 expressed in

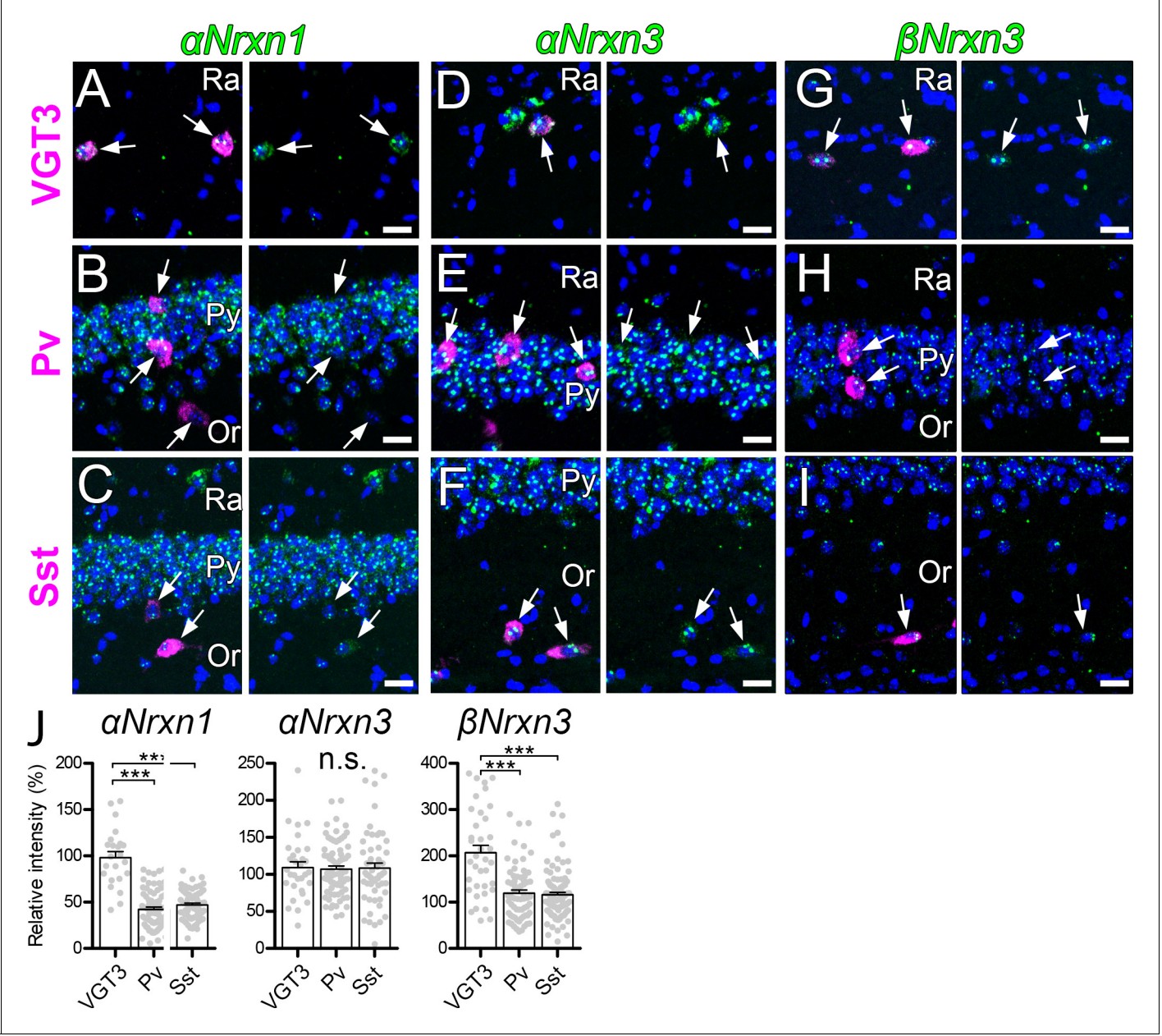

**Figure 6.** Expression of *Nrxn* isoforms in different types of inhibitory interneurons in the hippocampal CA1 region. Double FISH for *αNrxn1* (**A, B, and C**), *αNrxn3* (**D, E, and F**), and *βNrxn3* (**G, H, and I**) mRNAs in VGT3+ (**A, D, and G**), Pv+ (**B, E, and H**), and Sst+ (**C, F, and I**) in the hippocampus showing different levels of *Nrxn* mRNA (green) expression in different inhibitory interneurons (magenta). Note that the signal intensity in individual GABAergic neurons is variable, compared with that in glutamatergic neurons. Nuclei were stained with DAPI (blue). Or, st. oriens; Py, st. pyramidale; Ra, st. radiatum. (**J**) Summary scatterplots for *αNrxn1*, *αNrxn3,* and *βNrxn3* mRNAs in VGT3+, Pv+, and Sst+ inhibitory interneurons. Data are represented as mean ± SEM. ns, not significant; ***p<0.001 (Mann–Whitney U-test). Scale bars, 20 μm.
The online version of this article includes the following figure supplement(s) for figure 6:

**Figure supplement 1.** Expression of *Nrxn* isoforms in different types of inhibitory interneurons in the hippocampal CA1 region.

NrxnTKO/VGT3/RFP/tag-BFP neurons demonstrated reduced connectivity (***Figure 9D***), suggesting that βNrxn3-AS4 protein hinders synapse formation between VGT3/RFP terminals and postsynaptic CA1 pyramidal neurons.

Our rescue approach suggests that αNrxn+AS4 regulates inhibitory synaptic transmission with Nlgn3Δ. The structures of α and βNrxns are similar; therefore, different αNrxn+AS4 isoform(s) may

C

**Figure 7.** Transcriptomic similarity and endogenous *Nrxn* expression in hippocampal VGT3+, Pv+, and Sst+ inhibitory interneurons. (A) Single-cell t-SNE plot of 15 single cells obtained from VGT3+ (green: MU_VGT3.1–5), Pv+ (yellow: MU_Pv.1–5), and Sst+ (orange: MU_Sst.1–5) interneurons compared to Allen Brain Atlas single cells (CA1, cornu ammonis 1, pyramidal stratum, CA2; CA2, CA3; CA3, DG; dentate gyrus, L2 IT RHP; Layer2, intratelencephalic, retrohippocampal region, L2/3 IT CTX-2; Layer2/3, intratelencephalic, cortical region, L2/3 IT ENTl; Layer2/3, intratelencephalic, lateral entorhinal area, L5 IT TPE-ENT; Layer5, intratelencephalic, temporal, perirhinal, and ectorhinal area, L6 CT CTX; Layer6, near-projecting (NP), L6 corticothalamic (CT), Lamp5; Lamp5-positive, Meis2; Meis2-positive, NP SUB; near-projecting, subiculum, Pvalb; Pvalb-positive, Sncg; Sncg-positive, Sst; Sst-positive, SUB-ProS; subiculum-prosubiculum, Vip; Vip-positive). (B) Summary bar graph of *Nrxn* gene (*Nrxn1, 2,* and *3*) expression in VGT3+, Pv+, and Sst+ interneurons. (C–E) Summary graphs of splice isoforms of *Nrxn1* (C), *2* (D), and *3* (E) in VGT3+, Pv+, and Sst+ interneurons. *p<0.05, one-way ANOVA followed by Sidak's multiple comparisons test or Mann–Whitney U-test, N = 5 cells for each interneuron type.

*Figure 7 continued on next page*

*Figure 7 continued*

The online version of this article includes the following source data and figure supplement(s) for figure 7:

**Source data 1.** Inhibitory interneuron_tSNE data.
**Source data 2.** Heat map of inhibitory interneurons.
**Figure supplement 1.** Heat map of *Nrxn* gene expression in VGT3+, Pv+, and Sst+ inhibitory interneurons.

be able to functionally substitute αNrxn1+AS4. Although our FISH results demonstrated comparable levels of *αNrxn3* among VGT3+, Pv+, and Sst+ interneurons (**Figure 6D–F and J**), our single-cell RNA sequencing data indicate that *αNrxn3+AS4* is another dominant *Nrxn* splice isoform in VGT3+ interneurons (**Figure 7E**). Thus, we tested whether αNrxn3+/- AS4 functionally couples with Nlgn3Δ in NrxnTKO/VGT3/RFP slice cultures (**Figure 9E–H**). To our surprise, neither αNrxn3+AS4 nor αNrxn3-AS4 pairing with postsynaptic Nlgn3Δ had any effect on inhibitory synaptic transmission. These results strongly suggest that αNrxn1+AS4, but not αNrxn3+AS4, has a unique signal in the extracellular domain important for synaptic function with Nlgn3Δ.

## Discussion

Synaptic protein–protein interactions are critical for the development, maturation, and survival of neurons. However, it is technically challenging to physiologically characterize trans-synaptic CAM protein interactions in two different neurons due to the difficulty in identifying

**Table 1.** Nrxn transcript IDs used for quantification.

| Transcript ID | gene | α or β | AS4 plus minus |
|---|---|---|---|
| ENSMUST00000174331.7 | Nrxn1 | α | AS4- |
| ENSMUST00000054059.14 | Nrxn1 | α | AS4+ |
| ENSMUST00000072671.13 | Nrxn1 | α | AS4+ |
| ENSMUST00000160800.8 | Nrxn1 | α | AS4+ |
| ENSMUST00000160844.9 | Nrxn1 | α | AS4+ |
| ENSMUST00000161402.9 | Nrxn1 | α | AS4+ |
| ENSMUST00000159778.7 | Nrxn1 | β | AS4- |
| ENSMUST00000172466.7 | Nrxn1 | β | AS4+ |
| ENSMUST00000174337.7 | Nrxn1 | β | AS4+ |
| ENSMUST00000113461.7 | Nrxn2 | α | AS4- |
| ENSMUST00000137166.7 | Nrxn2 | α | AS4+ |
| ENSMUST00000235714.1 | Nrxn2 | α | AS4+ |
| ENSMUST00000077182.12 | Nrxn2 | α | AS4+ |
| ENSMUST00000113462.7 | Nrxn2 | α | AS4+ |
| ENSMUST00000236635.1 | Nrxn2 | α | AS4+ |
| TENSMUST00000113459.1 | Nrxn2 | β | AS4- |
| ENSMUST00000113458.7 | Nrxn2 | β | AS4+ |
| ENSMUST00000190626.6 | Nrxn3 | α | AS4- |
| ENSMUST00000057634.13 | Nrxn3 | α | AS4+ |
| ENSMUST00000163134.7 | Nrxn3 | α | AS4+ |
| ENSMUST00000167103.7 | Nrxn3 | α | AS4+ |
| ENSMUST00000167887.7 | Nrxn3 | α | AS4+ |
| ENSMUST00000110133.8 | Nrxn3 | β | AS4- |
| ENSMUST00000238943.1 | Nrxn3 | β | AS4- |
| ENSMUST00000110130.3 | Nrxn3 | β | AS4+ |

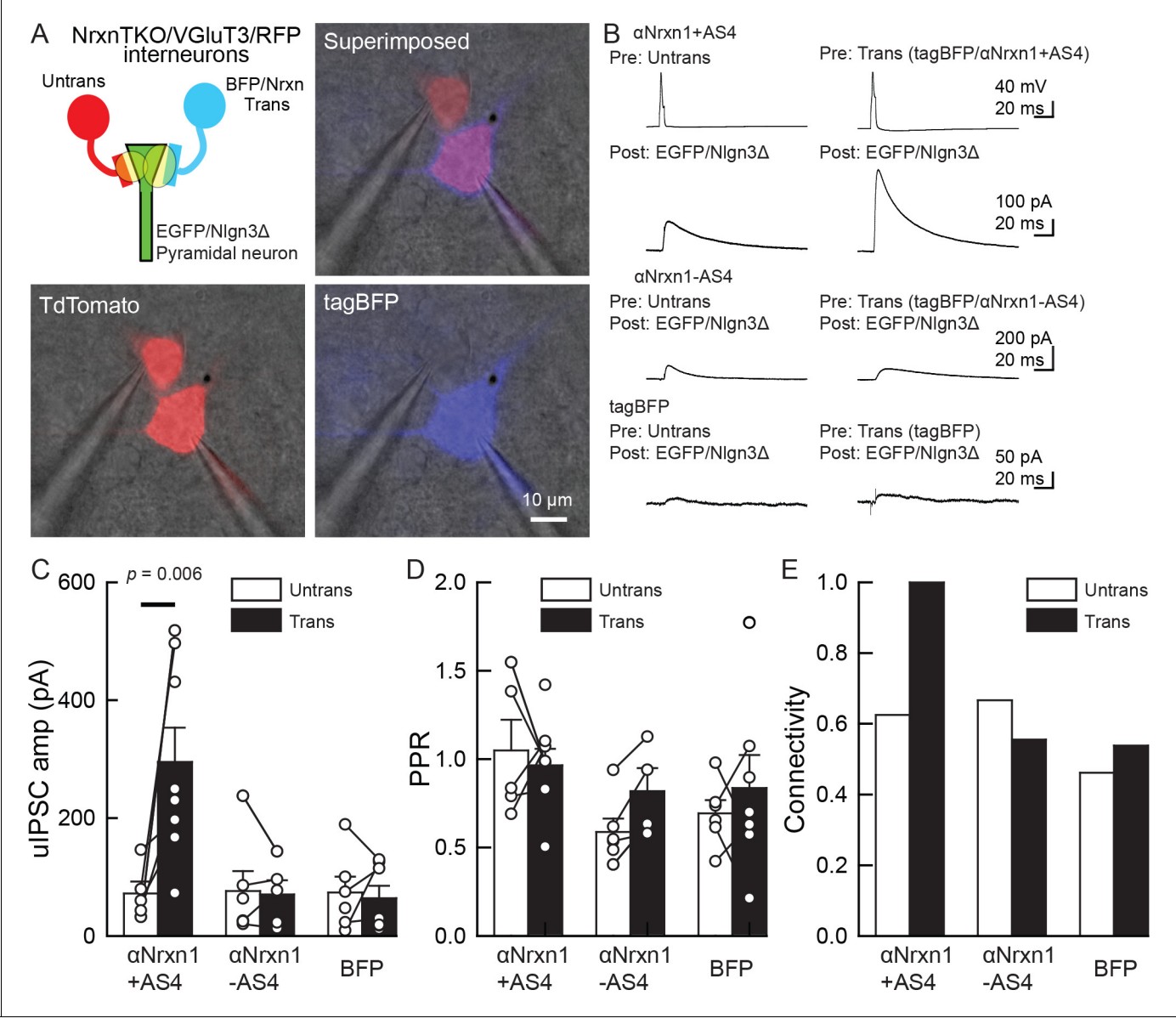

**Figure 8.** An αNrxn1+AS4–Nlgn3Δ synaptic *signal* enhances VGT3+ inhibitory synapse transmission. (A–C) Effect of pre- and postsynaptic overexpression of αNrxn1 and Nlgn3Δ, respectively, on unitary inhibitory synaptic transmission in organotypic slice cultures prepared from VGT3/NrxnTKO/RFP mice. (A) Configuration of triple whole-cell recording (left top), superimposed fluorescent, and Nomarski images (right top). (Bottom) Individual fluorescence and Nomarski images of TdTomato (left) and tag-blue fluorescent protein (tag-BFP; right). (B) Averaged sample unitary inhibitory postsynaptic current (uIPSC) traces. Nrxns and Nlgn3Δ were transfected in TdTomato-positive and CA1 pyramidal neurons, respectively, by electroporation. (C, D, and E) Summary of uIPSC amplitude (C), paired-pulse ratio (D), and connectivity (E). Numbers of cell pairs: αNrxn1+AS4, αNrxn1-AS4, and BFP at VGT3+ synapses (seven pairs/three mice, 9/4 and 9/4). The number of tested slice cultures is the same as that of cell pairs. Mann–Whitney U-test.

synaptically connected neuronal pairs in the brain. Co-culture approaches consisting of non-neuronal cells transfected with different *Nrxn* splice isoforms and dissociated neurons expressing endogenous Nlgns (or expressing Nrxn-binding proteins in non-neuronal cells and observing their interactions with endogenous Nrxns) have begun to elucidate the roles of trans-synaptic Nrxn/Nlgn isoforms on the clustering of pre-/postsynaptic molecules (*Chih et al., 2006*; *Kang et al., 2008*; *Ko et al., 2009*; *Nam and Chen, 2005*; *Scheiffele et al., 2000*). However, this approach is limited to primary cultures and cannot address whether these trans-synaptic interactions are sufficient to induce functional synapse diversification. To fully understand the physiological roles of trans-synaptic molecules, one

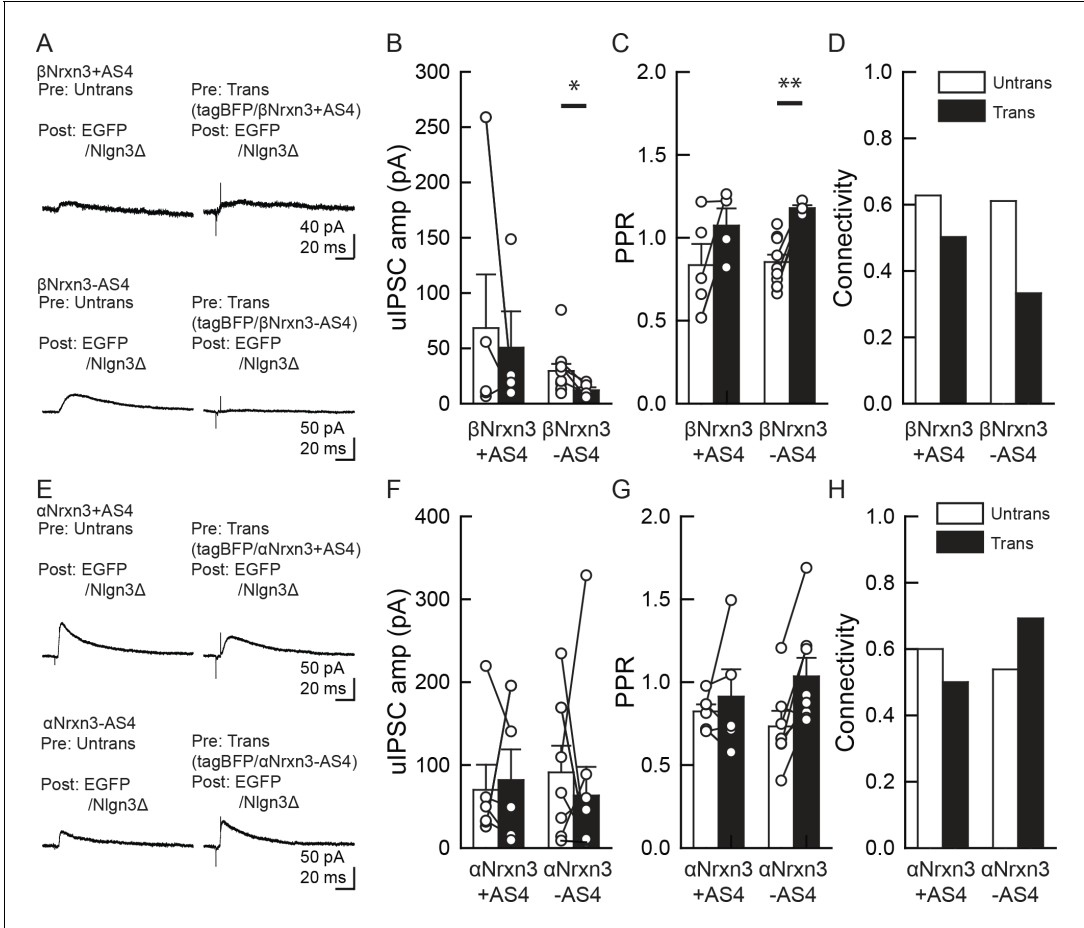

**Figure 9.** α/βNrxn3–Nlgn3Δ synaptic signals do not regulate VGT3+ inhibitory synapse transmission. **(A–D)** Effect of pre- and postsynaptic overexpression (OE) of βNrxn3(±AS4) and Nlgn3Δ, respectively, on unitary inhibitory synaptic transmission in organotypic slice cultures prepared from NrxnTKO/VGT3/RFP mice. **(E–H)** Effect of pre- and postsynaptic OE of αNrxn3(±AS4) and Nlgn3Δ, respectively, on unitary inhibitory synaptic transmission. **(A and E)** Averaged sample unitary inhibitory postsynaptic current (uIPSC) traces. Summary of uIPSC amplitude **(B and F)**, paired-pulse ratio **(C and G)**, and connectivity **(D and H)**. Numbers of cell pairs: βNrxn3+AS4, βNrxn3-AS4, αNrxn3+AS4, and αNrxn3-AS4 at VGT3+ synapses (eight pairs/three mice, 9/3, 5/3, and 11/3). The number of tested slice cultures is the same as that of cell pairs. Mann–Whitney U-test.

must be able to manipulate the expression of these molecules in pre- and postsynaptic neurons simultaneously followed by determination of the functional consequences of such manipulation. Using our newly developed gene electroporation method that enables us to transfect genes in minor cell types such as specific inhibitory interneurons (*Keener et al., 2020a*; *Keener et al., 2020b*), we demonstrated for the first time that αNrxn1+AS4 and Nlgn3Δ, which are endogenously expressed in VGT3+ inhibitory and CA1 pyramidal neurons (*Figure 7D*; *Futai et al., 2007*; *Shipman et al., 2011*; *Uchigashima et al., 2020*), respectively, form a specific signal that dictates inhibitory synaptic transmission.

It has been reported that Nlgn proteins regulate inhibitory synaptic transmission in an input cell-specific manner. For example, a Nlgn2 KO mouse line displays deficits in fast-spiking but not Sst+ interneuron-mediated inhibitory synaptic transmission in the somatosensory cortex (*Gibson et al., 2009*). Furthermore, Nlgn3 KO mice and the Nlgn3 R451C knock-in mutant line, which mimics a human autism mutation, showed Pv+ or Cck+ input-specific abnormal inhibitory synaptic transmission in the hippocampal CA1 region (*Földy et al., 2013*). Therefore, the function of postsynaptic Nlgns is determined by the type of presynaptic inputs it receives, supporting the intriguing hypothesis that specific Nrxn–Nlgn binding regulates synaptic function. Our gene expression and functional assays highlight that *αNrxn1* is abundantly expressed in VGT3+ interneurons compared with other interneurons (*Figure 6J*) and *αNrxn1+AS4* is the dominant *αNrxn1* splice isoform expressed in VGT3

+ interneurons (*Figure 7C*) which regulates inhibitory synaptic transmission with Nlgn3Δ (*Figure 8C*). Our FISH experiments, which did not distinguish the AS4 insertion, detected αNrxn1 at low levels in Pv+ interneurons (*Figure 6J*). The expression of *αNrxn1+AS4* was previously confirmed in Pv+ interneurons (*Fuccillo et al., 2015*). Considering no effects of Nlgn3Δ OEs on Pv+ inhibitory synapses (*Figure 2*), other postsynaptic mechanism(s) might exist to regulate inhibitory synaptic transmission with αNrxn1+AS4 at Pv+ inhibitory synapses.

Biochemical studies have demonstrated that any Nrxn can bind to any Nlgn with different affinities, with the exception of αNrxn1 and Nlgn1, which do not interact (*Boucard et al., 2005*; *Reissner et al., 2008*). The AS4 insertion has a critical role in changing Nrxns' affinity to postsynaptic binding partners, including Nlgn (*Südhof, 2017*). A splice insertion at AS4 in βNrxn1 can weaken its interaction with Nlgns (*Koehnke et al., 2010*). Indeed, we have reported that βNrxn1-AS4 but not βNrxn1+AS4 increases synaptic transmission through its interaction with Nlgn1 (*Futai et al., 2013*). Similarly, αNrxn1-AS4 specifically interacts with Nlgn2 and enhances inhibitory synaptic transmission (*Futai et al., 2013*). In contrast, we found that an insertion at AS4 in αNrxn1 can increase inhibitory synaptic transmission at VGT3+ synapses through a trans-synaptic interaction with Nlgn3Δ. It is particularly interesting that only αNrxn1+AS4, but not αNrxn3+AS4, encoded uIPSC enhancement with postsynaptic Nlgn3Δ (*Figures 8C* and *9F*). These results suggest that structural differences beyond the AS4 site exist between these two αNrxns. Since the structures of the major Nrxn domains, such as LNS and EGFA-EGFC, are similar between αNrxns, differential alternative splicing events that occur at other AS sites may regulate binding with Nlgn3Δ. Further structural and functional analyses targeting these AS events could reveal a novel AS structure that modulates Nrxn–Nlgn signaling on synaptic function. In contrast to biochemical analyses using purified Nrxn and Nlgn proteins, our results are based on intact synapses composed of a number of molecules including Nrxn and Nlgn proteins. Additional synaptic molecules could be involved in inhibitory synaptic functions mediated by trans-synaptic interactions between αNrxn1+AS4 and Nlgn3Δ at VGT3+ synapses. In particular, it will be important to address whether αNrxn1+AS1, αNrxn3+AS4, and βNrxn3-AS4, which are highly expressed in VGT3+ neurons (*Figure 7*), can encode specific synaptic functions when interacting with other postsynaptic Nrxn-binding partners such as Nlgn2, CST-3, and IgSF21 (*Pettem et al., 2013*; *Tanabe et al., 2017*; *Um et al., 2014*).

Nlgns function as homomeric or heteromeric dimers that bind to monomeric Nrxns (*Budreck and Scheiffele, 2007*; *Poulopoulos et al., 2012*). The function of human Nlgn3A2 at inhibitory synapses requires the presence of Nlgn2 in hippocampal neurons (*Nguyen et al., 2016*), suggesting that heterodimers of Nlgn3A2 and Nlgn2 are formed at inhibitory synapses. However, our finding of Nlgn3-mediated synaptic potentiation at VGT3+ synapses was based on Nlgn3Δ OE without the replacement of other Nlgns. Further studies are necessary to determine whether the function of Nlgn3Δ requires the formation of heterodimers with Nlgn2 at VGT3+ synapses.

Deletion of Nrxns or Nlgns has been reported to have little effect on synapse formation (*Chanda et al., 2017*; *Chen et al., 2017*; *Varoqueaux et al., 2006*). Our manipulation of the expression levels of either presynaptic Nrxns or postsynaptic Nlgn3 consistently did not affect synaptic connectivity, a measurement of the number of active synapses (*Figures 2*, *3*, and *6*). However, Nlgn3Δ formed new synapses only when αNrxn1+AS4 was simultaneously expressed in VGT3+ interneurons (*Figure 8E*). This suggests that trans-synaptic interactions of Nrxns and Nlgns may control not only synapse function but also synapse number, while the lack of Nrxns or Nlgns on either side of the synapse can be compensated by other CAM interactions. In this context, it is important to address whether the subcellular localization of Nrxn and Nlgn proteins is regulated by synaptic activity, such as homeostatic synaptic plasticity (*Mao et al., 2018*).

Note that our results, demonstrating the role of postsynaptic Nlgn3Δ and A2 isoforms in CA1 pyramidal neurons on inhibitory synaptic transmission, have some inconsistencies with a previously published study, which found that Nlgn3A2 differentially regulates Pv+ and Sst+ inhibitory synapses (*Horn and Nicoll, 2018*). Horn and Nicoll reported that OE of human Nlgn3A2 reduced Pv+ and increased Sst+ inhibitory synaptic transmission. The former finding is consistent with our results in *Figure 3B* and *Figure 3—figure supplement 1*, displaying that Nlgn3Δ or A2 OE reduce Pv+ uIPSC amplitude. This may suggest that Pv+ neurons have a unique trans-synaptic regulatory mechanism compared with other interneurons and that Nlgn3Δ or A2 OE may disrupt endogenous GABA$_A$R complexes at Pv+ inhibitory synapses. In contrast, Nlgn3Δ or A2 OE did not increase Sst+ uIPSCs (*Figure 3F* and *Figure 3—figure supplement 1*), as observed in *Horn and Nicoll, 2018*. This

difference might be due to variations in experimental approaches including the Nlgn3 clone used (human Nlgn3A2 versus mouse Nlgn3Δ splice isoform) and the duration of transgene or shRNA expression in hippocampal CA1 pyramidal neurons (2–3 weeks of OE versus 2–3 days of OE). Additionally, Horn and Nicoll crossed Pv-Cre and Sst-Cre, the same *cre* lines we tested, with Ai32 mice, a *cre*-dependent channelrhodopsin line (*Madisen et al., 2012*), to evoke Pv+ or Sst+ neuron-mediated synaptic transmission, respectively (*Horn and Nicoll, 2018*). However, our current and previous studies for mouse line validation indicate that while Pv-Cre and Sst-Cre lines exhibit highly specific *cre* expression in these cell types, these lines have leaky *cre* expression in other cell type(s) (*Figure 2—figure supplement 1*, yellow arrow heads) (*Mao et al., 2015*). Therefore, light-evoked activation of nonspecific cell types may contribute to the inconsistent results in synaptic transmission. Surprisingly, Nlgn3A2 did not increase VGT3+ inhibitory synaptic transmission. One possible explanation is that Nlgn3A2 couples with Cck+ neurons that do not express VGT3. Cck- and vasoactive intestinal polypeptide (VIP)-positive but VGT3-negative inhibitory interneurons have been identified in the hippocampal CA1 region (*Somogyi et al., 2004*). Furthermore, most of the Cck+ neurons in this area express CB1 transcripts (*Katona et al., 1999*). Therefore, CB1 signals in *Figure 1C* can originate from Cck+/VGT3+ and Cck+/VIP+ interneurons. Further studies are required to identify other types of interneurons that are capable of coupling with Nlgn3A2 by using other cell type-specific *cre* lines (e.g. VIP-*cre*).

Mutations and deletions in *Nrxn* loci are associated with neuropsychiatric and neurodevelopmental disorders. Copy number alterations (*Sebat et al., 2007*; *Szatmari et al., 2007*) and deleterious (*Yan et al., 2008*; *Zahir et al., 2008*) mutations in αNrxn1 are the most commonly reported Nrxn isoform-specific modifications predisposing people to epilepsy, autism spectrum disorders (ASDs), attention deficit hyperactivity disorder, intellectual disability (ID), schizophrenia (SCZ), and Tourette syndrome (*Ching et al., 2010*; *Clarke et al., 2012*; *Kim et al., 2008*; *Møller et al., 2013*). Rare mutations in Nlgn3 have been reported in patients with ID, SCZ, and ASDs (*Jamain et al., 2003*; *Parente et al., 2017*; *Yan et al., 2005*). Interestingly, ASD patients show abnormalities in memory discrimination (*Beversdorf et al., 2000*), which is partly mediated by the activity of hippocampal Cck+ interneurons (*Sun et al., 2020*; *Whissell et al., 2019*). This autistic phenotype may be caused by abnormal trans-synaptic interactions of αNrxn1 and Nlgn3 at VGT3+ synapses. In addition to our findings here, both Nrxn1 and Nlgn3 are expressed at other synapses, including excitatory synapses (*Baudouin et al., 2012*; *Budreck and Scheiffele, 2007*; *Uchigashima et al., 2019*; *Uchigashima et al., 2020*). It will be interesting to identify Nrxn1–Nlgn3 signals at different synapses to better understand the function of these genes on cognitive behavior.

## Materials and methods

### Animal and organotypic slice culture preparation

All animal protocols were approved by the Institutional Animal Care and Use Committee (IACUC) at the University of Massachusetts Medical School and Hokkaido University. Organotypic hippocampal slice cultures were prepared from postnatal 6- to 7-day-old mice of either sex, as described previously (*Stoppini et al., 1991*). Mice were WT (C57BL/6J, Jax #000664) expressing interneuron-specific TdTomato (Sst/RFP, Pv/RFP and VGT3/RFP), generated by crossing a TdTomato reporter line (Jax #007905) with Sst-Cre (Sst$^{Cre}$: Jax #013044), Pv-Cre (Pvalb$^{Cre}$: Jax #017320 or #008069), or VGT3-Cre (Slc17a8$^{Cre}$: Jax #018147) lines. The Nrxn1/2/3$^{f/f}$ mouse line was generated recently (*Uemura and Suzuki, 2020*; *Uemura et al., 2017*). The Nlgn3 KO mouse line was a gift from Dr. Kenji Tanaka (*Tanaka et al., 2010*).

### DNA and shRNA constructs

EGFP (Clontech), tag-BFP (Evrogen), *Nlgn3Δ, Nlgn3A2, αNrxn1 ± AS4, αNrxn3 ± AS4*, and *βNrxn3 ± AS4* genes were subcloned into a pCAG vector. The full AS configuration of the *αNrxn1* and *αNrxn3* clones were *αNrxn1(+AS1, -AS2, +AS3, ±AS4, +AS5, and -AS6)* and *αNrxn3(-AS1, -AS2, +AS3, ±AS4, -AS5, and -AS6)*. Two shRNA clones against Nlgn3, shNlgn3#1 (TRCN0000031940) and shNlgn3#2 (TRCN0000031939), were obtained from the RNAi Consortium (http://www.broad.mit.edu/genome_bio/trc/). The mouse *αNrxn3* clone was a gift from Dr. Ann Marie Craig (*Kang et al., 2008*).

## Antibodies

Primary antibodies raised against the following molecules were used: goat VGT3 (AB_2571854) (*Somogyi et al., 2004*), rabbit and goat VIAAT (RRID:AB_2571623 and AB_2571622) (*Fukudome et al., 2004*), rabbit CB1 (RRID:AB_2571591) (*Fukudome et al., 2004*), guinea pig Nlgn3 (RRID:AB_2571814) (*Uchigashima et al., 2019*; *Uchigashima et al., 2020*). rabbit Pv (RRID: AB_2571613, Sigma: P3088) (*Nakamura et al., 2004*), rabbit Sst (Penisula lab.: T-4103.0050), and goat/rabbit RFP (Rockland: 200-101-379 and 600-401-379, respectively).

## Single- and double-labeled FISH

Single/double FISH was performed using our recently established protocol (*Uchigashima et al., 2019*). VGT3+, Sst+, and Pv+ interneurons were identified with cRNA probes characterized previously (*Omiya et al., 2015*; *Song et al., 2014*; *Yamasaki et al., 2016*). All procedures were performed at room temperature unless otherwise noted. Briefly, fresh frozen sections were fixed with 4% paraformaldehyde, 0.1M PB for 30 min, acetylated with 0.25% acetic anhydride in 0.1M triethanolamine-HCl (pH 8.0) for 10 min, and prehybridized with hybridization buffer for 30 min. Hybridization was performed with a mixture of fluorescein- (1:1000) or DIG- (1:10,000) labeled cRNA probes in hybridization buffer overnight followed by post-hybridization washing using saline-sodium citrate buffers at 75°C. Signals were visualized using a two-step detection method. Sections were pretreated with DIG blocking solution for 30 min and 0.5% tryamide signal amplification (TSA) blocking reagent in Tris-NaCl-Tween 20 (TNT) buffer for 30 min before the first and second steps. During the first step, sections were incubated with peroxidase-conjugated anti-fluorescein antibody (1:500, Roche Diagnostics) and TSA Plus Fluorescein amplification kit (PerkinElmer) for 1 hr and 10 min, respectively. In the second step, sections were incubated with peroxidase-conjugated anti-DIG antibody (1:500, Roche Diagnostics) and TSA Plus Cy3 amplification kit (second step) with the same incubation times. Between the first and second steps, residual peroxidase activity was inactivated with 3% $H_2O_2$ in TNT buffer for 30 min. Sections were incubated with DAPI for 10 min for nuclear counterstaining (1:5000, Sigma-Aldrich).

## Western blotting

shRNA constructs were validated by western blotting. Nlgn- and shRNA-transfected HEK293T cells were solubilized in lysis buffer (10 mM Tris, pH 8.0, 200 mM NaCl, 1% Triton X-100, 1% SDS, and protease inhibitors) and loaded onto 8% SDS-PAGE gels. HEK293T cell line (Sigma) was authenticated by STR-PCR. Primary antibodies (1:1000 to 1:3000 dilution) were applied in blocking buffer (20 mM Tris, pH 7.4, 137 mM NaCl, 0.1% Tween 20, 1% bovine serum albumin, and 5% nonfat milk) for 2 hr at room temperature. Secondary antibodies were used at 1:2000 dilution. The signal was detected using an ECL detection kit (PerkinElmer Life Sciences).

## Immunohistochemistry

### Validation of *VGT3/RFP, Sst/RFP,* and *Pv/RFP* mouse lines

PFA-fixed brains (two brains for each line) were sliced (40 μm) and subjected to double staining with RFP (Rockland 200-101-379 or 600-401-379: 1/2000) and VGT3 (1 μg/ml) (*Somogyi et al., 2004*), Sst (Penisula lab., T-4103.0050: 1/2000) or Pv (Sigma: P3088, 1/2000).

### Triple staining of *Nlgn3* and synaptic markers

Mice were fixed by transcardial perfusion with 3% glyoxal fixative (*Richter et al., 2018*). Brains were cryoprotected with 30% sucrose in 0.1 M PB to prepare 50-μm-thick cryosections on a cryostat (CM1900; Leica Microsystems). All immunohistochemical incubations were performed at room temperature. Sections were incubated with 10% normal donkey serum for 20 min, a mixture of primary antibodies overnight (1 μg/ml), and a mixture of Alexa 488-, Cy3-, or Alexa 647-labeled species-specific secondary antibodies for 2 hr at a dilution of 1:200 (Invitrogen; Jackson ImmunoResearch, West Grove, PA). Images were taken with a confocal laser scanning microscope equipped with 473, 559, and 647 nm diode laser lines, and UPlanSApo (10×/0.40), UPlanSApo (20×/0.75), and PlanApoN (60×/1.4, oil immersion) objective lenses (FV1200; Olympus, Tokyo, Japan). To avoid crosstalk between multiple fluorophores, Alexa488, Cy3, and Alexa647 fluorescent signals were acquired sequentially using the 488 nm, 543 nm, and 633 nm excitation laser lines. All images show single

optical sections (800 × 800 pixels). The analysis was performed using ImageJ software (https://imagej.nih.gov/ij/). Briefly, the signal intensity and co-localization frequency of Nlgn3 puncta in the hippocampus CA1 were measured in the region of interest (ROI) selected from inhibitory synapses co-labeled for VIAAT and interneuron markers: VGT3, CB1, Pv, and Sst. To assess the noise levels for intensities and co-localizations, we analyzed images with the Nlgn3 channel rotated 90° to identify true close appositions of Nlgn3 signals and synaptic markers (*Singh et al., 2016*; *Stogsdill et al., 2017*). The noise levels for the signal intensity and co-localization frequency of Nlgn3 signals were comparable among the four distinct synapses (*Figure 1G*), suggesting that the distribution pattern or ROI of Nlgn3 signals and synaptic markers were unlikely to be biased. All the data for each group were obtained from two mice and pooled together.

## Single-cell sequencing and analysis

### Single-cell RNA extraction

The cytosol of four VGT3-positive neurons in CA1 origins was harvested using the whole-cell patch-clamp technique described previously (*Futai et al., 2013*; *Uchigashima et al., 2020*). The cDNA libraries were prepared using a SMART-Seq HT Kit (TAKARA Bio) and a Nextera XT DNA Library Prep Kit (Illumina) as per the manufacturers' instructions. The final product was assessed for its size distribution and concentration using a BioAnalyzer High Sensitivity DNA Kit (Agilent Technologies) and loaded onto an S1 flow cell on an Illumina NovaSeq 6000 (Illumina) and run for 2 × 50 cycles according to the manufacturer's instructions. De-multiplexed and filtered reads were aligned to the mouse reference genome (GRCm38) using HISAT2 (version 2.1.0) applying `-no-mixed` and `-no-discordant` options. Read counts were calculated using HTSeq by supplementing Ensembl gene annotation (GRCm38.78). Gene expression values were calculated as transcripts per million (TPM) using custom R scripts (*Source code 1*). Genes with no detected TPM in all samples were filtered out. Our data set was then combined with the 'Mouse Whole Cortex and Hippocampus SMART-seq' data portal from the Allen Institute for Brain Science where a complementary set of 76,533 total cells were primarily collected from >20 areas of mouse cortex and hippocampus of ~8-week-old pan-GABAergic, pan-glutamatergic, and pan-neuronal transgenic lines (*Yao et al., 2020*). The gene expression data matrix (matrix.csv) which stores raw read counts for every cell in the data set and cell metadata (metadata.csv) containing information such as sample names, brain regions of origin, cell type designations (e.g. 'GABAergic', 'Non-neuronal', and 'Glutamatergic') and cell type subclass designations (e.g. 'SST', 'L6 CT', and 'Astrocyte') was downloaded from the portal. Neuronal cells only from the hippocampus (2367 cells) were merged with our dataset (total 2382 cells). In order to minimize batch effect between our data and the Allen Brain Atlas, systematic differences in sequencing coverage across batches were removed by rescaling the size factors using the multiBatchNorm function from the batchelor R package (*Haghverdi et al., 2018*), and then a batch effect correction based on linear regression model was applied using the rescaleBatches function from the batchelor package. A tSNE plot was then generated using Rtsne R package (*van der Maaten and Hinton, 2008*). Eighty-five randomly selected hippocampal GABAergic neurons from the Allen Brain Atlas dataset and 15 of our cells were selected, and the batch-corrected expression levels of Nrxn genes were visualized in a heatmap using ComplexHeatmap R package (*Gu et al., 2016*). For splice isoform quantification, kallisto (*Bray et al., 2016*) was used by supplementing the transcript fasta file (Mus_musculus.GRCm38.cdna.all.fa). Each isoform was summarized manually to account for inclusion or exclusion of the AS4 exon in the α or β isoforms. The manually curated transcript IDs are provided in *Table 1*.

## Single-cell RT-qPCR

Isolation of single-cell cytosol and preparation of single-cell cDNA libraries were performed by the same method described in single-cell sequencing and analysis. For validation of the Nrxn KO mouse line, the following TaqMan gene expression assays (Applied Biosystems) were used: *Nrxn1* (Mm03808857_m1), *Nrxn2* (Mm01236856_m1), *Nrxn3* (Mm00553213_m1), and *Gapdh* (Mm99999915_g1). The relative expression of *Nrxns* was calculated as follows: Relative expression = $2^{Ct,Gapdh}/2^{Ct,Nrxns}$; Ct, threshold cycle for target gene amplification.

## Single-cell electroporation

A detailed protocol is described in our recent publication (*Keener et al., 2020a*; *Keener et al., 2020b*). Briefly, the slice cultures were perfused with filter-sterilized aCSF consisting of (in mM): 119 NaCl, 2.5 KCl, 0.5 $CaCl_2$, 5 $MgCl_2$, 26 $NaHCO_3$, 1 $NaH_2PO_4$, 11 glucose, and 0.001 mM tetrodotoxin (TTX, Hello Bio Inc), gassed with 5% $CO_2$/95% $O_2$, pH 7.4. Patch pipettes (4.5–8.0 MΩ) were each filled with plasmids containing either tag-BFP and *Nrxn* or EGFP and *Nlgn3Δ* (0.05 μg/μl for each plasmid) and respectively electroporated in TdTomato-positive VGT3+ interneurons and CA1 pyramidal neurons. The same internal solution for single-cell sequencing was used. A single electrical pulse train (amplitude: −5 V, square pulse, train: 500 ms, frequency: 50 Hz, pulse width: 500 μs) was applied to the target neurons. After electroporation, the electrode was gently retracted from the cell and the slices were transferred to a culture insert (Millipore) with slice culture medium in a petri dish and incubated in a 5% $CO_2$ incubator at 35°C for 3 days.

## Electrophysiology

Whole-cell voltage- and current-clamp recordings were performed on postsynaptic and presynaptic neurons, respectively. *Nlgn3Δ* and *Nlgn3A2* constructs or shRNAs were transfected at DIV6–9 or DIV2 and subjected to recordings at 2–3 or 5–12 days after transfection, respectively. DIV10–14 organotypic slice cultures prepared from WT (VGT3/RFP) and KO (NrxnTKO/VGT3/RFP) mice were evaluated for KO of Nrxns at VGT3+ synapses. The extracellular solution for recording consisted of (in mM): 119 NaCl, 2.5 KCl, 4 $CaCl_2$, 4 $MgCl_2$, 26 $NaHCO_3$, 1 $NaH_2PO_4$, 11 glucose, and 1 kynurenic acid (Sigma), gassed with 5% $CO_2$ and 95% $O_2$, pH 7.4. Thick-walled borosilicate glass pipettes were pulled to a resistance of 2.5–4.5 MΩ. Whole-cell voltage clamp recordings were performed with internal solution containing (in mM): 115 cesium methanesulfonate, 20 CsCl, 10 HEPES, 2.5 $MgCl_2$, 4 ATP disodium salt, 0.4 guanosine triphosphate trisodium salt, 10 sodium phosphocreatine, and 0.6 EGTA, adjusted to pH 7.25 with CsOH. For current-clamp recordings, cesium in the internal solution was substituted with potassium and the pH was adjusted with KOH. $GABA_A$ receptor-mediated inhibitory postsynaptic currents (IPSCs) were measured at Vhold ± 0 or −70 mV. Thirty to forty consecutive stable postsynaptic currents were evoked at 0.1 Hz by injecting current (1 nA) in presynaptic interneurons. Synaptic connectivity was tested by applying 25 consecutively paired (at 50 ms intervals) stimulations; responses larger than 5 pA observed within 5 ms after the onset of either of the pulses were counted as evoked unitary $GABA_A$R-IPSC. Recordings were performed using a Multi-Clamp 700B amplifier and Digidata 1440, digitized at 10 kHz and filtered at 4 kHz with a low-pass filter. Data were acquired and analyzed using pClamp (Molecular Devices).

## Statistical analyses

Results are reported as mean ± SEM. Statistical significance, set at $p < 0.05$, was evaluated by one- or two-way ANOVA with Sidak's post hoc test for multiple comparison, Mann-Whitney U-test, and Student's *t*-test for two-group comparison.

## Acknowledgements

This work was supported by grants from the National Institutes of Health Grants (R01NS085215 to KF, T32 GM107000 and F30MH122146 to AC), the Global Collaborative Research Project of Brain Research Institute, Niigata University (G2905 to KF), Grants-in-Aid for Scientific Research (19100005 to MW; 15K06732 and 20H03349 to MU), and The Naito Foundation (to MU). The authors thank Ms. Naoe Watanabe and Ms. Rie Natsume for skillful technical assistance. We thank Dr. Paul Gardner for comments on an earlier draft of the manuscript.

## Additional information

### Funding

| Funder | Grant reference number | Author |
| --- | --- | --- |
| National Institute of Neurological Disorders and Stroke | R01NS085215 | Kensuke Futai |

| National Institute of Mental Health | F30MH122146 | Amy Cheung |
|---|---|---|
| Japan Society for the Promotion of Science | 19100005 | Masahiko Watanabe |
| Japan Society for the Promotion of Science | 15K06732 | Motokazu Uchigashima |
| Japan Society for the Promotion of Science | 20H03349 | Motokazu Uchigashima |
| National Institute of General Medical Sciences | T32 GM107000 | Amy Cheung |
| Niigata University | G2905 | Kensuke Futai |
| Naito Foundation | | Motokazu Uchigashima |
| Riccio Fund for Neuroscience | | Kensuke Futai |
| Whitehall Foundation | | Kensuke Futai |

The funders had no role in study design, data collection and interpretation, or the decision to submit the work for publication.

## Author contributions

Motokazu Uchigashima, Conceptualization, Data curation, Software, Formal analysis, Funding acquisition, Validation, Investigation, Methodology, Writing - original draft; Kohtarou Konno, Data curation, Formal analysis, Investigation, Visualization, Methodology, Writing - original draft; Emily Demchak, Timmy Le, Investigation; Amy Cheung, Data curation, Formal analysis, Funding acquisition, Investigation, Writing - original draft; Takuya Watanabe, Data curation, Formal analysis, Investigation, Writing - original draft; David G Keener, Formal analysis, Investigation, Writing - original draft; Manabu Abe, Data curation, Investigation, Methodology, Project administration; Kenji Sakimura, Supervision, Validation, Writing - original draft; Toshikuni Sasaoka, Writing - original draft; Takeshi Uemura, Resources, Investigation, Writing - original draft; Yuka Imamura Kawasawa, Resources, Data curation, Software, Formal analysis, Investigation, Visualization, Methodology, Writing - original draft; Masahiko Watanabe, Resources, Supervision, Funding acquisition, Investigation, Writing - original draft; Kensuke Futai, Conceptualization, Resources, Data curation, Formal analysis, Supervision, Funding acquisition, Validation, Investigation, Visualization, Methodology, Writing - original draft, Project administration

## Author ORCIDs

Motokazu Uchigashima (iD) https://orcid.org/0000-0002-0878-2233
Amy Cheung (iD) https://orcid.org/0000-0002-4708-0293
Masahiko Watanabe (iD) https://orcid.org/0000-0001-5037-7138
Kensuke Futai (iD) https://orcid.org/0000-0002-3433-3407

## Ethics

Animal experimentation: This study was performed in strict accordance with the recommendations in the Guide for the Care and Use of Laboratory Animals of the National Institutes of Health. Animal protocol (#A2208) was approved by the Institutional Animal Care and Use Committee (IACUC) at the University of Massachusetts Medical School.

## Decision letter and Author response

Decision letter https://doi.org/10.7554/eLife.59545.sa1
Author response https://doi.org/10.7554/eLife.59545.sa2

# Additional files

## Supplementary files

• Source code 1. tpm calculation.

• Transparent reporting form

## Data availability

Sequencing data have been deposited in GEO under accession code GSE150989.

The following dataset was generated:

| Author(s) | Year | Dataset title | Dataset URL | Database and Identifier |
|---|---|---|---|---|
| Uchigashima M, Konno K, Demchak E, Cheung A, Watanabe T, Keener D, Abe M, Le T, Sakimura K, Sasaoka T, Uemura T, Kawasawa YI, Watanabe M, Futai K | 2020 | Specific Neuroligin3-$\alpha$ Neurexin1 Signaling Regulates GABAergic Synaptic Function in Mouse Hippocampus | https://www.ncbi.nlm. nih.gov/geo/query/acc. cgi?acc=GSE150989 | NCBI Gene Expression Omnibus, GSE150989 |

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

# Appendix

**Appendix 1—key resources table**

| Reagent type (species) or resource | Designation | Source or reference | Identifiers | Additional information |
|---|---|---|---|---|
| Cell line (Human) | HEK 293T | Sigma | | Authenticated with STR profiling, tested negative for mycoplasma |
| Antibody | Anti-RFP (goat polyclonal) | Rockland | Cat#: 200-101-379 (RRID: AB_2744552) | IF (1:2000) |
| Antibody | Anti-RFP (rabbit polyclonal) | Rockland | Cat#: 600-401-379 (RRID: AB_2209751) | IF (1:2000) |
| Antibody | Anti-PV (mouse monoclonal) | Sigma | Cat#: P3088 (RRID:AB_477329) | IF (1:1000) |
| Antibody | Anti-PV (rabbit polyclonal) | Frontier Institute | Cat#: PV-Rb-Af750 (RRID: AB_2571613) | IF (1 µg/ml) |
| Antibody | Anti-Sst antibody (rabbit polyclonal) | Peninsula Lab | Cat#: T-41003.0050 | IF (1:2000) |
| Antibody | Anti-VIAAT (goat polyclonal) | Frontier Institute | Cat#: PV-Rb-Af750 (RRID: AB_2571613) | IF (1 µg/ml) |
| Antibody | Anti-VIAAT (rabbit polyclonal) | Frontier Institute | Cat#: VGAT-Rb-Af500 (RRID:AB_2571622) | IF (1 µg/ml) |
| Antibody | Anti-NL3 (rabbit polyclonal) | Frontier Institute | Cat#: Nlgn3-Gp-Af880 (RRID:AB_2571814) | IF (1 µg/ml) |
| Antibody | Anti-VGluT3 (goat polyclonal) | Frontier Institute | Cat#: VGluT3-Go-Af870 (RRID:AB_2571854) | IF (1 µg/ml) |
| Antibody | Anti-CB1 (rabbit polyclonal) | Frontier Institute | Cat#: CB1-Rb-Af380 (RRID:AB_2571591) | IF (1 µg/ml) |
| Antibody | Cy3-AffiniPure Donkey anti-Guinea pig IgG | Jackson ImmunoResearch | Cat# 706-165-148 (RRID: AB_2340460) | IF (1:200) |
| Antibody | Donkey anti-Rabbit IgG, Alexa Fluor 488 | Invitrogen | Cat#: A-21206 (RRID:AB_2535792) | IF (1:200) |
| Antibody | Donkey anti-Goat IgG, Alexa Fluor 488 | Invitrogen | Cat#: A-11055 (RRID:AB_2534102) | IF (1:200) |
| Antibody | Donkey anti-Goat IgG, Alexa Fluor 647 | Invitrogen | Cat#: A-21447 (RRID:AB_2535864) | IF (1:200) |
| Antibody | Donkey anti-Rabbit IgG, Alexa Fluor 647 | Invitrogen | Cat#: A-31573 (RRID:AB_2536183) | IF (1:200) |
| Antibody | Anti-Fluorescein-POD, Fab fragments from sheep | Roche | Cat#: 11426346910 (RRID:AB_840257) | FISH (1:500) |
| Antibody | Anti-Digoxigenin-POD, Fab fragments from sheep | Roche | Cat#: 11207733910 (RRID:AB_840257) | FISH (1:500) |
| Chemical compound, drug | tRNA from brewer's yeast | Roche | Cat#:10109517001 | FISH |

*Continued on next page*

*Appendix 1—key resources table continued*

| Reagent type (species) or resource | Designation | Source or reference | Identifiers | Additional information |
|---|---|---|---|---|
| Chemical compound, drug | Sheep serum | Sigma-Aldrich | Cat#: S3772 | FISH |
| Chemical compound, drug | DAPI | Sigma-Aldrich | Cat#: D9542 | Nuclear staining (1:5000) |
| Chemical compound, drug | Kynurenic acid | Sigma-Aldrich | Cat#: K3375 | Electrophysiology |
| Recombinant DNA reagent | pCAG-HA-Nlgn3Δ (plasmid) | *Uchigashima et al., 2020* | | Subclone HA- Nlgn3Δ in pCAG vector |
| Recombinant DNA reagent | pCAG-HA-Nlgn3A2 (plasmid) | *Uchigashima et al., 2020* | | Subclone HA- Nlgn3A2 in pCAG vector |
| Recombinant DNA reagent | pCAG-Tag-BFP (plasmid) | This paper | Tag-BFP (Evrogen) | Subclone Tag-BFP in pCAG vector |
| Recombinant DNA reagent | pCAG-EGFP (plasmid) | *Futai et al., 2013* | EGFP (Clontech) | Subclone EGFP in pCAG vector |
| Recombinant DNA reagent | pCAG-HA-αNrxn1 + AS4 (plasmid) | This paper | | Subclone HA-αNrxn1 + AS4 in pCAG vector |
| Recombinant DNA reagent | pCAG-HA-αNrxn1 – AS4 (plasmid) | *Futai et al., 2013* | | Subclone HA-αNrxn1 – AS4 in pCAG vector |
| Recombinant DNA reagent | pCAG-HA-αNrxn3 + AS4 (plasmid) | This paper | | Subclone HA-αNrxn3 + AS4 in pCAG vector |
| Recombinant DNA reagent | pCAG-HA-αNrxn3 – AS4 (plasmid) | *Kang et al., 2008* | | Subclone HA-αNrxn3 – AS4 in pCAG vector |
| Recombinant DNA reagent | pCAG-HA-βNrxn3 + AS4 (plasmid) | This paper | | Subclone HA-βNrxn3 + AS4 in pCAG vector |
| Recombinant DNA reagent | pCAG-HA-βNrxn3 – AS4 (plasmid) | This paper | | Subclone HA-βNrxn3 – AS4 in pCAG vector |
| Recombinant DNA reagent | shNlgn3#1 | Sigma Aldrich | TRCN0000031940 | shRNA in pLKO.1 vector |
| Recombinant DNA reagent | shNlgn3#2 | Sigma Aldrich | TRCN0000031939 | shRNA in pLKO.1 vector |
| Recombinant DNA reagent | pBS SK (-)-α Nrxn1#1 (plasmid) | *Uchigashima et al., 2019* | | Subclone *αNrxn1* (186–631, NM_020252.3) in pBS SK (-) vector |
| Recombinant DNA reagent | pBS SK (-)-α Nrxn1#2 (plasmid) | *Uchigashima et al., 2019* | | Subclone *αNrxn1* (631–1132, NM_020252.3) in pBS SK (-) vector |
| Recombinant DNA reagent | pBS SK (-)-β Nrxn1#1 (plasmid) | *Uchigashima et al., 2019* | | Subclone *βNrxn1* (379–824, XM_006523818.2) in pBS SK (-) vector |
| Recombinant DNA reagent | pBS SK (-)-β Nrxn1#2 (plasmid) | *Uchigashima et al., 2019* | | Subclone *βNrxn1* (882–1490, XM_006523818.2) in pBS SK (-) vector |
| Recombinant DNA reagent | pBS SK (-)-α Nrxn2#1 (plasmid) | *Uchigashima et al., 2019* | | Subclone *αNrxn2* (120–370, NM_001205234) in pBS SK (-) vector |
| Recombinant DNA reagent | pBS SK (-)-α Nrxn2#2 (plasmid) | *Uchigashima et al., 2019* | | Subclone *αNrxn2* (894–1209, NM_001205234) in pBS SK (-) vector |

*Continued on next page*

*Appendix 1—key resources table continued*

| Reagent type (species) or resource | Designation | Source or reference | Identifiers | Additional information |
|---|---|---|---|---|
| Recombinant DNA reagent | pBS SK (-)-β Nrxn2#1 (plasmid) | *Uchigashima et al., 2019* | | Subclone *βNrxn2* (188–375, AK163904.1) in pBS SK (-) vector |
| Recombinant DNA reagent | pBS SK (-)-β Nrxn2#2 (plasmid) | *Uchigashima et al., 2019* | | Subclone *βNrxn2* (543–714, AK163904.1) in pBS SK (-) vector |
| Recombinant DNA reagent | pBS SK (-)-α Nrxn3#1 (plasmid) | *Uchigashima et al., 2019* | | Subclone *αNrxn3* (207–795, NM_001198587) in pBS SK (-) vector |
| Recombinant DNA reagent | pBS SK (-)-α Nrxn3#2 (plasmid) | *Uchigashima et al., 2019* | | Subclone *αNrxn3* (796–1782, NM_001198587) in pBS SK (-) vector |
| Recombinant DNA reagent | pBS SK (-)-β Nrxn3#1 (plasmid) | *Uchigashima et al., 2019* | | Subclone *βNrxn3* (85–530, NM_001252074) in pBS SK (-) vector |
| Recombinant DNA reagent | pBS SK (-)-β Nrxn3#2 (plasmid) | *Uchigashima et al., 2019* | | Subclone *βNrxn3* (771–1235, NM_001252074) in pBS SK (-) vector |
| Recombinant DNA reagent | pBS SK (+)-VGT3 (plasmid) | *Omiya et al., 2015* | | Subclone *VGluT3* (22–945, NM_182959) in pBS SK (+) vector |
| Recombinant DNA reagent | pBS SK (+)-Pv (plasmid) | *Yamasaki et al., 2016* | | Subclone *Parv* (57–389, NM_013645) in pBS SK (+) vector |
| Recombinant DNA reagent | pBS SK (+)-Sst (plasmid) | *Song et al., 2014* | | Subclone *Sst* (133–408, NM_012659) in pBS SK (+) vector |
| Commercial assay or kit | Fluorescein RNA Labeling Mix | Roche | Cat#: 11685619910 | cRNA probe synthesis kit |
| Commercial assay or kit | DIG RNA Labeling Mix | Roche | Cat#: 11277073910 | cRNA probe synthesis kit |
| Commercial assay or kit | T3 RNA Polymerase | Promega | Cat#: P2083 | cRNA probe synthesis |
| Commercial assay or kit | T7 RNA Polymerase | Promega | Cat#: P2075 | cRNA probe synthesis |
| Commercial assay or kit | Blocking reagent | Roche | Cat#: 11096176001 | FISH |
| Commercial assay or kit | TSA Blocking reagent | PerkinElmer | Cat#: FP1020 | FISH |
| Commercial assay or kit | TSA Plus Cyanine three and Fluorescein System | PerkinElmer | Cat#: NEL753001KT | FISH |
| Commercial assay or kit | SMRT-Seq HT Kit | Takara Bio | Cat# 64437 | RNA-seq library prep kit |
| Commercial assay or kit | Nextera XT DNA Library Preparation Kit | Illumina | FC-131–1096 | RNA-seq library prep kit |
| Commercial assay or kit | Nextera XT Index Kit v2 Set A | Illumina | FC-131–2001 | RNA-seq library prep kit |
| Commercial assay or kit | NovaSeq 6000 S1 Reagent Kit (100 cycles) | Illumina | 20012865 | Illumina sequencing reagent |

*Continued on next page*

*Appendix 1—key resources table continued*

| Reagent type (species) or resource | Designation | Source or reference | Identifiers | Additional information |
|---|---|---|---|---|
| Software, algorithm | BBTools | DOE Joint Genome Institute | https://jgi.doe.gov/data-and-tools/bbtools/ (RRID:SCR_016968) | RNA-seq read filtering and trimming |
| Software, algorithm | HISAT2 | Center for Computational Biology at Johns Hopkins University | https://ccb.jhu.edu/software/hisat2/manual.shtml (RRID:SCR_015530) | RNA-seq read alignment |
| Software, algorithm | HTSeq | Simon Anders and Fabio Zanini | https://htseq.readthedocs.io/en/master/overview.html (RRID:SCR_005514) | RNA-seq read quantification |
| Software, algorithm | kallisto | Lior Pachter lab | https://pachterlab.github.io/kallisto/about (RRID:SCR_016582) | RNA-seq pseudoalignment |
| Software, algorithm | R | The R Foundation | https://www.r-project.org/ | Bioinformatic data manipulation, calculation and graphical display |
| Software, algorithm | MetaMorph | Molecular devices | https://www.moleculardevices.com/?_ga = 2.213249172.1046710400.1598576508–1999962342.1597912565 (RRID:SCR_002368) | Intensity and count analysis |
| Software, algorithm | ImageJ | | https://imagej.nih.gov/ij/ (RRID:SCR_003070) | Intensity and count analysis |
| Software, algorithm | pCLAMP 10.6 | Molecular devices | RRID:SCR_011323 | Electrophysiological data analysis |

