## [Decision Letter]

**Acceptance summary:**

This study demonstrates that a particular type of inhibitory synapse in the hippocampus is regulated by the signaling between presynaptic neurexin and postsynaptic neuroligin adhesion molecule variants.

**Decision letter after peer review:**

Thank you for submitting your article "A Specific Neuroligin3-αNeurexin1 Code Regulates GABAergic Synaptic Function in Mouse Hippocampus" for consideration by *eLife*. Your article has been reviewed by three peer reviewers, and the evaluation has been overseen by a Reviewing Editor and Gary Westbrook as the Senior Editor. The following individual involved in review of your submission has agreed to reveal their identity: Markus Missler (Reviewer #3). The reviewers have discussed the reviews with one another and the Reviewing Editor has drafted this decision to help you prepare a revised submission for *eLife*.

As the editors have judged that your manuscript is of interest, but as described below additional experiments are required before it is published. However, we would like to draw your attention to changes in our revision policy that we have made in response to COVID-19 (https://elifesciences.org/articles/57162). Because many researchers have temporarily lost access to the labs, we will give authors as much time as they need to submit revised manuscripts.

Summary:

This study by Uchigashima et al. investigates to what extent Neurexin-Neuroligin interactions define synapse functions in an inhibitory microcircuit in the hippocampal CA1. The authors propose that Neuroligin-3 and Neurexin-1α regulate inhibitory synaptic transmission at synapses formed by vGluT3-positive (VGT3+) interneurons on CA1 hippocampal pyramidal neurons. This is based on (1) immunohistochemical data that localize Nlgn3 to synapses of VGT3+ interneurons, (2) the regulation of unitary inhibitory postsynaptic currents by Nlgn3 overexpression or knockdown in postsynaptic pyramidal neurons in organotypic slices from VGT3+ reporter mice, and (3) the finding that Nrxn deletion in VGT3+ interneurons prevents the effect of Nlgn3 overexpression in postsynaptic neurons. Single-cell RNA sequencing and in situ hybridization are presented to show that Nrxn1α and Nrxn3β mRNAs are prominent Nrxn isoforms in VGT3+ interneurons, and Nrxn1α SS4 rescued Nrxn deletion effects. If amended by critical controls and a more careful interpretation of the presumed mechanism, the manuscript would make a highly interesting contribution to the field of synapse specification by synaptic cell adhesion molecules.

Essential revisions:

1) The role of neurexins for transmission at the VGT3+ interneuron-to-CA1-pyramidal cell synapse remains unclear: The authors claim that Nrxn are important for the transmission at the VGT3+ synapse. However, I do not see the necessary experiment to substantiate such a general claim, for example, by comparing VGT3+synapses of control/undeleted to deleted NrxnTKO slices. Figure 5 rather shows that Nrxn is required to mediate the effect of overexpression of transfected Nlgn3Δ in CA1 neurons but this might be due to the overexpression itself. Thus, this effect would be more convincing if lack of Nrxn at VGT3+ synapses caused the opposite result on uIPSCs.

2) The physiological findings are based on paired recordings where genetically labeled VGT3+ interneurons are stimulated. These cells are sparse and heterogeneously distributed in CA1 (Pelkey et al., 2017). Given the issues with Cre driver lines, a more thorough analysis is needed to establish that bona fide VGT3+ interneurons contribute to the reported findings. The scattered distribution of the individual RFP+ cells in single-cell RNAseq data (Figure 7A) adds to this concern about cell identity. The only relevant evidence presented is the IHC analysis in Figure 2—figure supplement 1A-C but it does not include probes for other interneuron types in the CA1 that shows the specificity of the VGT3+ label. The authors should provide evidence that the Cre driver line used here indeed selectively labels VGT3+ interneurons.

3) To actually prove the specificity of the impact of Nlgn3Δ splice variant on inhibitory transmission from VGT3+ interneurons (Figure 2), an important control is missing: Another Nlgn3 variant, in which the A inserts are present, should be tested in the overexpression experiment. I do acknowledge that the authors compared different Nlgn3 variants in a recent paper (Uchigashima et al., 2020) in a related setting but no data exist for the proposed specificity of the Nlgn3Δ splice variant at VGT3+ synapses as far as I can see. Along this line, the authors see an increase in IPSCs when o/e NLGN3 in pyramidal neurons when Sst+ neurons were stimulated using optogenetics, whereas Horn and Nicoll did not see any changes. Horn and Nicoll used NLGN3A2 (including A2 insert) and in this study the construct doesn't have A1 or A2 insert. Perhaps they can discuss if the A2 insert can potentially be the culprit for the discrepancy if this is a potential confounding factor.

---

## [Author Response]

Essential revisions:1) The role of neurexins for transmission at the VGT3+ interneuron-to-CA1-pyramidal cell synapse remains unclear: The authors claim that Nrxn are important for the transmission at the VGT3+ synapse. However, I do not see the necessary experiment to substantiate such a general claim, for example, by comparing VGT3+synapses of control/undeleted to deleted NrxnTKO slices. Figure 5 rather shows that Nrxn is required to mediate the effect of overexpression of transfected Nlgn3Δ in CA1 neurons but this might be due to the overexpression itself. Thus, this effect would be more convincing if lack of Nrxn at VGT3+ synapses caused the opposite result on uIPSCs.

We appreciate this comment, which highlights an important point. To address this major concern, we performed dual pre- and postsynaptic recordings on VGT3+ interneurons and neighboring pyramidal neurons, respectively, in organotypic slice cultures prepared from VGT3/RFP (wild-type) and NrxnTKO/VGT3/RFP mice. Importantly, we found that the absence of Nrxns in VGT3+ neurons reduced uIPSC amplitude, PPR and connectivity. These findings suggest that Nrxns regulate inhibitory synaptic transmission at VGT3+ inhibitory synapses and are presented in Figure 5B-E.

2) The physiological findings are based on paired recordings where genetically labeled VGT3+ interneurons are stimulated. These cells are sparse and heterogeneously distributed in CA1 (Pelkey et al., 2017). Given the issues with Cre driver lines, a more thorough analysis is needed to establish that bona fide VGT3+ interneurons contribute to the reported findings. The scattered distribution of the individual RFP+ cells in single-cell RNAseq data (Figure 7A) adds to this concern about cell identity. The only relevant evidence presented is the IHC analysis in Figure 2—figure supplement 1A-C but it does not include probes for other interneuron types in the CA1 that shows the specificity of the VGT3+ label. The authors should provide evidence that the Cre driver line used here indeed selectively labels VGT3+ interneurons.

We performed additional validation studies on the VGT3/RFP mouse line. Using immunohistochemistry, we tested the expression of RFP (TdTomato) in Pv+ and Sst+ interneurons in the hippocampal CA1 region. Anti-Pv or -Sst antibody staining displayed clear segregation of RFP and Pv or Sst signals, indicating specific expression of RFP in VGT3+, but not Pv+ or Sst+, inhibitory interneurons. These results are presented in Figure 2—figure supplement 1D, E and G.

3) To actually prove the specificity of the impact of Nlgn3Δ splice variant on inhibitory transmission from VGT3+ interneurons (Figure 2), an important control is missing: Another Nlgn3 variant, in which the A inserts are present, should be tested in the overexpression experiment. I do acknowledge that the authors compared different Nlgn3 variants in a recent paper (Uchigashima et al., 2020) in a related setting but no data exist for the proposed specificity of the Nlgn3Δ splice variant at VGT3+ synapses as far as I can see. Along this line, the authors see an increase in IPSCs when o/e NLGN3 in pyramidal neurons when Sst+ neurons were stimulated using optogenetics, whereas Horn and Nicoll did not see any changes. Horn and Nicoll used NLGN3A2 (including A2 insert) and in this study the construct doesn't have A1 or A2 insert. Perhaps they can discuss if the A2 insert can potentially be the culprit for the discrepancy if this is a potential confounding factor.

This is an interesting point. Horn and Nicoll presented that overexpression of human Nlgn3A2 increased Sst+ inhibitory synaptic transmission. We assessed the effect of mouse Nlgn3A2 overexpression on unitary inhibitory synaptic transmission mediated by VGT3+, Pv+ and Sst+ inhibitory interneurons. To our surprise, we found that neither VGT3+ nor Sst+ inhibitory synapses increased uIPSC in neurons overexpressing Nlgn3A2 compared with untransfected neurons. Nlgn3A2 OE reduced uIPSC at Pv+ inhibitory synapses, which was also demonstrated in Horn and Nicoll’s study. These findings suggest that Nlgn3A2 may differentially regulate inhibitory synapses depending on the interneuron type and are presented in Figure 3—figure supplement 1. We expanded the Discussion section to further elaborate the inconsistency between our and Horn and Nicoll’s results. We consider that the inconsistent results of Nlgn3A2 OE in Sst+ synapses can arise from variations in experimental approaches, including the Nlgn3 clone tested [human (Horn and Nicoll, 2018) versus mouse (this study)], the duration of transgene OE (two to three weeks of OE versus two to three days of OE), age of organotypic slice cultures tested (14-21 DIV versus 7-14 DIV) and the method used to evoke synaptic current (optogenetics versus direct recording from presynaptic neurons).

One possible mechanism to explain the lack of Nlgn3A2 OE effect on VGT3+ synapses is that Nlgn3A2 couples with Cck+ neurons that do not express VGT3. It has been reported that there are VGT3 negative Cck+ interneurons that express Vasoactive Intestinal Polypeptide (VIP), another Cck marker, in the hippocampal CA1 region (Somogyi et al., 2004). As Figure 1C indicates, Nlgn3 preferentially localizes at CB1+ inhibitory terminals. Most of the Cck+ neurons in this area express CB1 transcripts (Katona et al., 1999; Pelkey et al., 2017). Therefore, our CB1 staining in Figure 1C can be derived from Cck+/VGT3+ and Cck+/VIP+ interneurons. It will be important to test the effect of Nlgn3A2 OE on VIP+ inhibitory synapses in future studies.